# Assessment of wood burning versus fossil fuel contribution to wintertime black carbon and carbon monoxide concentrations in Athens, Greece

Athina-Cerise Kalogridis[1], Stergios Vratolis[1], Eleni Liakakou[2], Evangelos Gerasopoulos[2], Nikolaos Mihalopoulos[2], Konstantinos Eleftheriadis[1]

[1]Institute of Nuclear & Radiological Sciences & Technology, Energy & Safety, National Centre of Scientific Research "Demokritos", Ag. Paraskevi, 15310, Greece

[2] Institute for Environmental Research and Sustainable Development, National Observatory of Athens, Metaxa & V. Pavlou, P. Penteli, 15236, Athens, Greece

*Correspondence to*: Athina-Cerise Kalogridis (akalogridi@ipta.demokritos.gr)

**Abstract**

The scope of this study was to estimate the contribution of fossil fuel and wood burning combustion to black carbon (BC) and carbon monoxide (CO) during wintertime, in Athens. For that purpose, in-situ measurements of equivalent Black Carbon (eBC) and CO were simultaneously conducted in a suburban and an urban background monitoring site in Athens during three months of winter 2014-2015. For the deconvolution of eBC into eBC emitted from fossil fuel ($BC_{ff}$) and wood burning ($BC_{wb}$), a method based on the spectral dependency of the absorption of pure black carbon and brown carbon was used. Thereafter, $BC_{wb}$ and $BC_{ff}$ estimated fractions were used along with measured CO concentrations in a multiple regression analysis, in order to quantify the contribution of each one of the combustion sources to the ambient CO levels. For a comparative analysis of the results, we additionally estimated the wood-burning and fossil fuel contribution to CO, calculated on the basis of their $CO/NO_x$ emission ratios. The results indicate that during wintertime BC and CO are mainly emitted by local sources within the Athens Metropolitan Area (AMA). Fossil fuel combustion, mainly from road traffic, is found to be the major contributor to both eBC in $PM_{2.5}$ and CO ambient concentrations in AMA. However, wintertime wood burning makes a significant contribution to the observed eBC (of about 30%) and CO concentrations (on average, 11 % and 16 % of total CO in the suburban and urban background sites respectively). Both, BC and CO from biomass burning ($BC_{wb}$ and $CO_{wb}$, respectively) present a clear diurnal pattern with highest concentrations during night, supporting the local domestic heating as their main source.

## 1 Introduction

Air pollution, which is originating largely from combustion processes, is a very important environmental concern in Athens, like in other large urban agglomerations around the world. High population (3.75 million in the metropolitan area) and the confinement of commercial and industrial activities in a relatively small area (approximately 450 km$^2$), has led to severe environmental degradation. Over the years high loadings of atmospheric pollutants have been documented (Chaloulakou et al., 2005; Eleftheriadis et al., 1998, 2014; Kalabokas et al., 1999; Theodosi et al., 2011). Combustion processes used for transportation, power generation and other human activities produce a complex mixture of chemical pollutants (Cohen et al., 2004), which at any given location have characteristics depending on the relative contributions of the different sources of pollution and on the effects of the local geo-climatic factors. Black carbon (BC) aerosol and carbon monoxide (CO) are two major products of incomplete combustion and are important atmospheric components because of their substantial impact on health (Ostro et al., 2015), including respiratory and cardiovascular effects, as well as on climate (Zanatta et al., 2016). BC refers to the absorbing components of soot and is the second most significant contributor to climate change (Andreae and

Gelencsér, 2006; Bond et al., 2013). On the other hand, CO strongly influences the oxidative capacity of the atmosphere (by reacting with the OH radicals), and thereby alters the lifetime of methane and other greenhouse gases (Seinfeld and Pandis, 2012). The potential adverse health and climate effects associated with exposure to high levels of BC and CO, motivates a thorough characterization of their emission from different sources. In urban environments, fossil fuel combustion is the major source of BC and CO, mainly related to motor vehicle exhausts. However, biomass combustion, from forest fires (especially in summer; Diapouli et al., 2014), or from domestic heating (in winter) may also contribute significantly to their ambient levels. Improvement of air quality in Athens after measures adopted during the last decades are described by Kanakidou et al. (2011) and are in line with proposed mitigation strategies (Aleksandropoulou et al., 2012). Vrekoussis et al. (2013) used satellite observations of $NO_2$ and $SO_2$ over Athens, to show that the economic crisis resulted in acceleration of the reduction of air pollutants in Athens. In the recent years, a resurgence in the use of biofuels over more expensive fuels for heating has been observed in Europe (Denier van der Gon et al., 2015; Gonçalves et al., 2011), especially in Greece where the economic crisis has tripled the fossil fuel cost in a few years (Saffari et al., 2013). The technology of domestic wood burning in Athens is known to suffer from low burning efficiency. The extensive use of wood burning observed in recent years resulted in considerable emissions from incomplete combustion, i.e. CO, hydrocarbons and soot particles during cold months. Paraskevopoulou et al. (2015), reported the impact of wood combustion (dominant fuel for domestic heating) on air quality in Athens, as an almost 30 % increase in the contribution of particulate organic matter to the urban aerosol mass, from winter 2012 to winter 2013. At the same time, a long term analysis of EC concentrations in Athens (Paraskevopoulou et al., 2014) revealed a significant increase in wintertime EC in the period 2011-2013. A significant increase in winter evening CO level has been reported for the period 2012-2015 by Gratsea et al. (2017) and attributed to the increase in wood burning use. On the other hand, during summertime a consistent decrease is encountered in the last decade as a result of the simultaneous reduction in traffic and industrial activity due to the economic crisis in Greece (Diapouli et al., 2017a).

Even though biomass burning from domestic heating has been recognized as a main source of atmospheric pollutants in Southern Europe (Denier van der Gon et al., 2015; Giannoni et al., 2012; Gonçalves et al., 2011; Paglione et al., 2014; Saffari et al., 2013) emission estimates are still scarce and the associated uncertainty remains high. This is because wood consumption statistics are difficult to obtain since wood is often non-commercial and emission factors vary greatly with wood type, combustion equipment and flame temperature. As a matter of fact, reported emission factors of PM for different types of residential combustion appliances range between 10 and 2000 mg per MJ (Mega Joule) of fuel burnt (Kocbach Bølling et al., 2009). In a similar way, CO emission factors from fireplaces and traditional or eco-labelled woodstoves range typically from 30-120 g $kg^{-1}$ (ratio of the mass of CO emitted related to the mass of the burnt fuel) (AIRUSE, 2015). The spatial variability of the carbonaceous aerosol pollutants is also of great interest with respect to the health impacts of their major contributing sources in the urban $PM_{10}$.

In this context, the scope of the present study is to investigate, based on ambient measurements, the impact of biomass burning versus fossil fuel use on the air pollution observed in Athens during wintertime. For that purpose, three-months of continuous and simultaneous measurements of equivalent Black Carbon (eBC) by aethalometers and CO at a suburban and an urban background site of Athens were analyzed and compared. The measured eBC was deconvoluted into two fractions using a model based on the different spectral dependencies of light absorption by pure black carbon (related to fossil fuel) and brown carbon (linked to wood-burning emissions) (Sandradewi et al., 2008). For simplicity we refer to these light absorbing carbon fractions as "BC" with an additional index for specifying the wood burning ($BC_{wb}$) or fossil fuel ($BC_{ff}$) origin. A new approach based on the relations between CO, $BC_{wb}$ and $BC_{ff}$ was used for the source apportionment of CO, and compared with a method based on the CO/$NO_x$ emission ratios.

## 2 Material and Methods

### 2.1 Sampling Sites

Simultaneous measurements of eBC and CO were performed at the National Center of Scientific Research (NCSR) Demokritos (DEM) and at the National Observatory of Athens (NOA), from December 6$^{th}$ 2014 until March 10$^{th}$ 2015 (Figure 1). The campus of Demokritos is situated at the foot of Mount Hymmettus in Agia Paraskevi and covers an area of 600 acres in a forest of pine trees. Situated at 7 km NNE from Athens centre (Triantafyllou et al., 2016), the GAW-Demokritos measurement station (37.99 N, 23.82 E, 270 m a.s.l) is considered representative of the suburban areas of the Athens Metropolitan Area (AMA). NOA's station is located at its central premises at Thission, in the center of Athens and on the top of Nymphs Hill (38.0 N, 23.7 E, 107 m a.s.l.). Its central setting, still relatively far (further than 500 m) from main traffic lines, can be considered ideal for monitoring the air pollution urban background of Athens.

### 2.2 Measurements of aerosol light absorption and carbon monoxide

#### 2.2.1 Aerosol light absorption and equivalent black carbon

The aerosol light absorption coefficient, $b_{abs}$, was retrieved at each station by means of a 7-wavelength (370, 470, 520, 590, 660, 880 and 950 nm) aethalometer (Magee Scientific Corp., Berkeley, CA 94703, USA) with a 5-min temporal resolution. At DEM, the new generation AE33 aethalometer model was used, which provides a real-time compensation for multiple scattering in the filter matrix and loading effects using the DualSpot Technology® (Drinovec et al., 2015). The AE33 sampled aerosol through a PM$_{2.5}$ cut-off inlet. At NOA, a Portable Aethalometer® Model AE42 was used and aerosol sampling was performed with no size-cut. As BC mainly contributes to PM$_1$ (Laborde et al., 2013; Wang et al., 2015), differences in the eBC concentrations due to the different aerodynamic diameters of sampled aerosols are expected to be negligible. Raw absorption coefficients at a given wavelength $\lambda$ ($b_{aeth,\lambda}$) from AE42 dataset were corrected from loading and scattering effects following the procedure introduced by Weingartner et al. (2003).

$$b_{abs,\lambda} = \frac{b_{ATN,\lambda}}{C_0 \times R(ATN)_\lambda} \tag{1}$$

$$R(ATN)_\lambda = \left(\frac{1}{f_\lambda} - 1\right) \times \frac{\ln(ATN_\lambda) - \ln 10}{\ln 50 - \ln 10} + 1 \tag{2}$$

As described in Eq. (1), a $C_0$ constant was used to correct for multiple scattering by the filter fibers and the scattering of the aerosols embedded in the filter, whereas a $R(ATN)$ function enabled to compensate the loading effect, i.e. the linearity loss in the relationship between the transmission of light through the sample-laden filter and the amount of the light-absorbing sample on the filter (Eq. 2). The value of 3.5 was used for $C_0$ as recommended in Zanatta et al. (2016). The compensation parameter $f_\lambda$ is a parameter that mainly depends on the single scattering albedo of aerosol and is expressed as:

$$f_\lambda = \alpha \cdot (1 - SSA_\lambda) + 1 \tag{3}$$

where $SSA_\lambda$ is the aerosol single scattering albedo and $\alpha$ a constant parameter, varying in the range 0.82–0.88 for the different wavelengths (950–370 nm). Since no simultaneous scattering coefficient measurements were available during this campaign, $f_\lambda$ values ($f_{370}$=1.319, $f_{470}$=1.288, $f_{520}$=1.244, $f_{590}$=1.213, $f_{660}$=1.213, $f_{880}$=1.203, $f_{950}$=1.203) given in Drinovec et al. (2015) for an urban site with a single scattering albedo of about 0.75 were used for loading compensation at the urban background site. In order to estimate uncertainties related to the chosen $f_\lambda$ values, absorption coefficients calculated with $f_\lambda$ values taken from Drinovec et al. (2015) were compared with those using $f_\lambda$ values calculated for a $SSA$ value of 0.8. Differences in the absorption coefficients calculated using $f_\lambda$ values calculated for an $SSA$ of 0.8 (e.g. $f_6$=1.166) and those found in Drinovec et al., (2015) were found to be lower than 1%. The performance of the loading compensation algorithm was examined using the approach described in Drinovec et al., (2015), where the dependence of the absorption Ångström exponent α to the attenuation of light is analysed. The slope that derives from the regression between the Ångström exponent α (calculated from α=ln($b_{abs,470}$

$_{nm}/b_{abs,880nm})/\ln(880/470))$ and the attenuation has been calculated. The value of the slope is equal to -0.0055 before compensation and close to 0 (-0.0005) after compensation of the data, thus indicating that the shadowing effect was correctly accounted for.

Eventually, aerosol light absorption coefficients were converted into mass concentration of the equivalent BC (eBC) as defined by Petzold et al. (2013). BC is historically defined from aethalometer measurements at 880 nm. At 880 nm, no significant difference in mass absorption cross-section (MAC) at 880 nm between eBC originating from traffic or wood-burning emission is expected (Zotter et al., 2017). eBC mass concentration was derived in this study by multiplying the $b_{abs}$ coefficient at 880 nm with a constant value of MAC of 4.6 m$^2$ g$^{-1}$ (determined from the comparison with simultaneous measurements at DEM of elemental carbon). An extensive description of EC/OC measurements at DEM is available in Diapouli et al. (2014). Assuming an absorption Ångstrom exponent of 1.0, the MAC used here is 6.13 when adjusted to 637 nm. Our MAC value is at the lower limit of the values reported by Zanatta et al. (2016), for nine rural background stations across Europe (7.5-13.3 m$^2$ g$^{-1}$, calculated for 637 nm), and within the range of values reported by Hitzenberger et al. (2006) for an urban background site in Vienna (5.9 -7.5 at 637 nm).

### 2.2.2 Carbon monoxide and nitrogen oxides

Ambient CO mixing ratios were measured at DEM station at a time resolution of 1 Hz using a Cavity Ring-Down Spectroscopy analyzer (Model G2401, Picarro, CA, USA), which provides high resolution and low detection limit CO, $CO_2$ and $CH_4$ ambient mixing ratios in line with GAW standards. Air was pulled through a 5-m line at about 0.4 L min$^{-1}$, and water was removed from the sample using a Nafion$^{TM}$ copolymer membrane dryer (http://www.permapure.com/resources/all-about-nafion-and-faq/). CO measurements were obtained with a typical precision (one sigma) of about ±4 ppbv in a 1-s measurement for concentrations ranging between 75 and 300 ppbv. Additionally, hourly measurements of $NO_x$ were available from the monitoring station of the Greek Ministry of Environment and Energy (www.ypeka.gr) situated 300 m from DEM station.

CO and $NO_x$ were determined at the NOA station with a 1-min integration time using a Horiba APMA-360 series automatic gas analyser (NDIR technique, scale: 0-20 ppmv, lower detectable limit: 0.05 ppmv) and a Horiba APNA-360 series (chemiluminescence technique, scale: 0-1000 ppbv, lower detectable limit: 0.5 ppbv), respectively.

### 2.3 Source apportionment of black carbon from fossil fuel and wood burning combustion

Source apportionment of the ambient BC concentrations was based on the method developed by Sandradewi et al. (2008) and successfully applied in several studies (Favez et al., 2009; Fourtziou et al., 2017; Fuller et al., 2014; Petit et al., 2014; Sciare et al., 2011). This model relies on the first assumption that the total absorption at a wavelength $\lambda$, $b_{abs}(\lambda)$, is a combination of absorption due to fossil fuel ($b_{abs}(\lambda)_{ff}$) and wood burning ($b_{abs}(\lambda)_{wb}$) aerosols:

$$\boldsymbol{b_{abs}(\lambda)} = \boldsymbol{b_{abs}(\lambda)_{ff}} + \boldsymbol{b_{abs}(\lambda)_{wb}} , \tag{4}$$

Secondly, it is based on the difference in the dependency of the absorption coefficient on wavelength assuming that absorption from fossil fuel and biomass burning emissions follow different spectral dependencies. The wavelength ($\lambda$) dependent absorption of light by aerosols ($b_{abs}$) is proportional to $\lambda^{-\alpha}$ where α is the absorption Ångström exponent such that:

$$\frac{b_{abs}(\lambda_1)_{ff}}{b_{abs}(\lambda_2)_{ff}} = \left(\frac{\lambda_1}{\lambda_2}\right)^{-\alpha_{ff}} , \tag{5}$$

And respectively:

$$\frac{b_{abs}(\lambda_1)_{wb}}{b_{abs}(\lambda_2)_{wb}} = \left(\frac{\lambda_1}{\lambda_2}\right)^{-\alpha_{wb}} , \tag{6}$$

Light absorption measurements at $\lambda_1 = 470 \; nm \; (UV)$, and $\lambda_2 = 950 \; nm \; (IR)$ are used in this approach due to the fact that when compared to BC from fossil fuel combustion (BC$_{ff}$), wood burning aerosols (BC$_{wb}$) exhibit greater absorption in the near ultraviolet. This enhanced absorption at near UV for wood burning aerosols is due to the presence of absorbing organic molecules, especially polycyclic aromatic hydrocarbons and humic-like substances (Hoffer et al., 2006). Even though different

pairs of near-UV and near-IR wavelengths can be used, it is recommended to use the pair 470 nm versus 950 nm. The choice of 470 against 370 is even more critical as explained in Zotter et al (2017) since VOCs or other absorbing non-BC particles can interfere with measurements with the 370-nm aethalometer channel. By solving equations 4-6, unique values of $b_{abs}(\lambda_{UV})_{wb}$, $b_{abs}(\lambda_{IR})_{wb}$, $b_{abs}(\lambda_{UV})_{ff}$ and $b_{abs}(\lambda_{IR})_{ff}$, can be calculated, thus leading to the determination of $BC_{wb}$ and $BC_{ff}$:

$$b_{abs}(\lambda_{UV})_{wb} = \frac{1}{1-\left(\frac{\lambda_{UV}}{\lambda_{IR}}\right)^{-\alpha_{ff}}*\left(\frac{\lambda_{UV}}{\lambda_{IR}}\right)^{\alpha_{wb}}} \times \left( b_{abs}(\lambda_{UV}) - \left(\frac{\lambda_{UV}}{\lambda_{IR}}\right)^{-\alpha_{ff}} * b_{abs}(\lambda_{IR}) \right), \tag{7}$$

$$b_{abs}(\lambda_{IR})_{wb} = \left(\frac{\lambda_{UV}}{\lambda_{IR}}\right)^{\alpha_{wb}} * b_{abs}(\lambda_{UV})_{wb} , \tag{8}$$

$$b_{abs}(\lambda_{UV})_{ff} = b_{abs}(\lambda_{UV}) \ - b_{abs}(\lambda_{UV})_{wb}, \tag{9}$$

$$b_{abs}(\lambda_{IR})_{ff} = b_{abs}(\lambda_{IR}) \ - b_{abs}(\lambda_{IR})_{wb} , \tag{10}$$

$$BC_{ff} = \frac{b_{abs}(\lambda_{IR})_{ff}}{b_{abs}(\lambda_{IR})} \times EBC , \tag{11}$$

$$BC_{wb} = EBC - BC_{ff} , \tag{12}$$

The application of the model requires the selection of suitable Ångström exponents for fossil fuel ($\alpha_{ff}$) and wood burning ($\alpha_{wb}$), since one of the greatest uncertainties of the model is associated with the a priori assumed α values for both types of emissions. Reported Ångström exponents range between 0.8-1.1 for pure traffic. For wood burning a wider range of values has been observed (0.9-3.5), even though $\alpha_{wb}$ equal to 2.0 has long been considered a typical value for wood burning aerosol (Favez et

al., 2009; Fuller et al., 2014; Herich et al., 2011; Petit et al., 2014; Sciare et al., 2011). Recently, Zotter et al. (2017) recommended to use $\alpha_{ff}$=0.9 and $\alpha_{wb}$=1.68, obtained by fitting the model outputs (calculated with the absorption coefficients at 470 and 950 nm) against the fossil fraction of EC derived from [14]C measurements. At DEM site, a previous study showed that values of $a_{wb}$ below 1.7 were not appropriate for the specific site (Diapouli et al., 2017b). On top of that during fire events, Ångström exponent values up to 2 have been observed at DEM. Taking into consideration these results, absorption Ångström

exponents of 0.9 and 2.0 for pure traffic ($\alpha_{ff}$) and wood burning ($\alpha_{wb}$), respectively, were used in this study.

It should also be noted that coal-burning organic aerosol is known to significantly absorb light at near UV wavelengths (Yang et al., 2009) and may thus interfere with $b_{abs}(\lambda_{UV})_{wb}$. Lignite coal is the single most important local energy source in Greece (Kavouridis, 2008). However, interferences from coal use are expected to be very low, as the lignite-fired power plants are located far away from Athens (>200 km distance).

**2.4 Source apportionment of carbon monoxide from fossil fuel and wood burning combustion**

The partitioning of CO ambient concentrations into different sources has been investigated in a limited number of studies and was mainly based on the variable isotopic composition of CO (Gros et al., 2002; Kato et al., 1999; Saurer et al., 2009). In the absence of isotopic analysis, we use two different models for the source apportionment, based on the correlations between CO and other combustion tracers.

**2.4.1 Model 1: the CO-NO$_x$ linear model.**

The CO/NO$_x$ ratio has been used in the past as a diagnostic to characterize different type of emission sources (Fujita et al., 1992; Ravindra et al., 2006; Saurer et al., 2009; Wahlina et al., 2001). It can serve as a useful tool for source apportioning CO concentrations at monitoring stations part of national networks where only regulated air pollutants are often measured. The CO-NOx linear model, introduced by Saurer et al. (2009), relies on the fact that both CO and NO$_x$ are common products of combustion processes. Assuming that the only significant combustion processes in urban environments are traffic and wood burning for residential heating (in addition to the regional background, and a minor contribution by industrial processes), the concentrations of NO$_x$ and CO can be expressed as:

$$[NO_x] = [NO_x]_{bgd} + [NO_x]_{ff} + [NO_x]_{wb} , \tag{13}$$

$$[CO] = [CO]_{bgd} + [CO]_{ff} + [CO]_{wb} , \tag{14}$$

where *[X]$_{ff}$, [X]$_{wb}$* represent the concentration of the tracer *X* resulting from fossil fuel (mainly traffic) and wood-burning, respectively, whereas *[X]$_{bgd}$* represent the background concentration of *X*.

The CO-NO$_x$ linear model is based on the distinct CO/NO$_x$ ratios for the two emission sources, where the wood-burning emission ratio, *r$_{wb}$*, is much larger than the one for traffic, *r$_{ff}$*. Considering that photochemical processes do not substantially affect the ambient concentrations of CO and NO$_x$ in winter, the ratios of the concentrations can be regarded as approximately the same as their respective emission ratios, $[CO]_{ff}/[NOx]_{ff} \approx r_{ff}$ and $[CO]_{wb}/[NOx]_{wb} \approx r_{wb}$. Based on this assumption, we can consider that *r$_{ff}$* and *r$_{wb}$* are given, and as a consequence Eq. (13) can be rewritten as:

$$[NO_x] = [NO_x]_{bgd} + \frac{[CO]_{ff}}{r_{ff}} + \frac{[CO]_{wb}}{r_{wb}}, \tag{15}$$

Equations 14 and 15 allow *[CO]$_{ff}$* and *[CO]$_{wb}$* to be determined. The concentration of CO originating from wood burning emissions can be expressed as:

$$[CO]_{wb} = \frac{r_{wb}}{r_{ff} - r_{wb}} \times \left[ [CO]_{bgd} - [CO] + r_{ff}([NO_x] - [NO_x]_{bgd}) \right] , \tag{16}$$

Defining the emission ratios $r_{wb}$ and $r_{ff}$ is a crucial step for the source apportionment of CO. The methodology used for their selection is presented in section 3.3.2

It is important here to mention the limitations of this model for CO apportionment. Firstly, it requires an *a priori* knowledge of the emission ratios *r$_{ff}$* and *r$_{wb}$*. Secondly, it is based on the hypothesis that the CO/NO$_x$ ratio remains constant, while in fact it could be affected by photochemistry. CO is a long-lived species with an atmospheric lifetime of several days to several weeks; hence photochemical processes influence CO concentrations on a limited extent. In contrast, NO$_x$ are much more reactive species. Consequently, any change in reactive nitrogen compounds, mainly by photochemistry, would alter the CO/NO$_x$ ratio.

**2.4.2 Model 2: CO-BC$_{wb}$-BC$_{ff}$ multiple linear regression model**

The second model for CO source apportionment is based on the existing relation between the concentrations of CO and BC, introducing advantages in order to overcome the limitations of the previously presented CO-NO$_x$ linear model.

In a similar manner to model 1, considering both CO and BC are exclusively produced by combustion processes and that in the urban environment the CO/BC ratios can be regarded as equivalent to their source emission ratio, the CO concentration can be expressed as:

$$[CO] = [CO]_{bdg} + r'_{ff} \times [BC]_{ff} + r'_{wb} \times [BC]_{wb} \, , \tag{17}$$

where $r'_{ff} = [CO]_{ff}/[BC]_{ff}$, and $r'_{wb} = [CO]_{wb}/[BC]_{wb}$ are the relevant emission ratios at the source. The difference of our approach in this second model resides in the way that this equation is solved. Unlike the CO-NO$_x$ linear model, here *a priori* knowledge of $r_{wb}$ and $r_{ff}$ emission ratios is not required. Instead, BC$_{ff}$ and BC$_{wb}$ are known variables (determined previously using the method presented in section 2.2), and $r'_{ff}$ and $r'_{wb}$ can be calculated by a multiple linear regression model applied to Eq. (17).

Using $r'_{ff}$ and $r'_{wb}$ resulting from the model, the concentration of CO attributed to fossil fuel and wood burning sources can be estimated such that:

$$CO_{ff} = r'_{ff} \times [BC]_{ff} \, , \tag{18}$$

and,

$$CO_{wb} = r'_{wb} \times [BC]_{wb} \tag{19}$$

Moreover, the hypothesis of negligible photochemical chemistry is met for the time scale considered in this study and for the long-lived species BC and CO, and therefore should not have a significant impact on their ambient ratio.

While multiple linear regressions are known techniques for source apportionment, they have not yet been applied to investigate wood burning contribution to CO using the aethalometer's model results. The considerable increase in measurements carried out using aethalometers makes this technique an interesting and very useful methodology for apportioning CO concentrations.

## 3. Results and Discussion

### 3.1 Levels and diurnal variations of black carbon and carbon monoxide

Statistical summary of eBC and CO levels at NOA and DEM stations, as well as their respective time series are displayed in Table 1 and Fig. 2, respectively. Median eBC and CO levels were, respectively, 2.3 and 1.7 times higher at NOA station compared to DEM station. In particular, on days with stagnant atmospheric conditions (low wind speed), concentrations of both combustion tracers were up to 10 times higher at NOA compared to DEM. During days with more turbulence, the levels of eBC and CO were similar at both stations, as a result of intensive mixing and uniform horizontal pollutants' dispersion in the Athens valley. As shown in Fig. 3, eBC mass concentrations observed at NOA station are similar to previously reported values in various urban background sites of European highly populated cities, whereas eBC concentrations at DEM are of the same order of magnitude as in residential urban or suburban areas in Europe.

Diurnal cycles of eBC and CO, as well as wind speed and temperature have been calculated as 1-h mean values and are shown in Fig. 4. eBC and CO exhibit similar diurnal variability. At both stations, maximum concentrations of eBC and CO occur during morning hours (between 08:00-09:00 a.m) and late evening (between 08:00-09:00 p.m), suggesting common emission sources. Based on traffic volume data (Grivas et al., 2012), a first peak in the emissions from transportation is expected around 08:00 when people commute to work, followed by a plateau from 08:00-18:00, and a secondary peak until 21:00, after when traffic is decreasing. Wood burning emissions from residential heating are expected to increase during the evening, when temperatures drop and people are back home. As a result, the first peak in eBC and CO concentrations occurs during morning traffic rush hours, while the surface boundary layer is still shallow. At NOA peaks are quite more pronounced compared to DEM, suggesting that each site is under the influence of different small scale dynamics in the Athens valley (Tombrou et al., 2007). It is also interesting to observe that the average minimum of concentrations at NOA occurs during midday due to a higher boundary layer height (BLH) and corresponding aerosol dilution during daytime, when both sampling sites are under the same well-mixed atmosphere, winds are stronger, and consequently the pollutants are more homogeneously dispersed in the metropolitan area. The second peak at NOA during night-time is the result of the Nocturnal Boundary Layer (NBL) formation, with the site (elevation 107 m) well within the nocturnal boundary layer (Kassomenos and Koletsis, 2005), leading to an

accumulation of atmospheric pollutants from combustion sources active at night. During the late hours of the night, the minimum of the 24-h concentrations are observed at the periphery of the basin (DEM), where during stagnant conditions advection from the urban pollution sources is reduced. Occasional downslope winds (katabatic winds) from Hymettus Mountain (Amanatidis et al., 1992) may enhance air exchange from outside the nocturnal boundary layer (NBL), at the same time induce

a build up, and an increase in concentrations at the Athens Basin.

## 3.2 Source apportionment of BC and diurnal variability

Following the deconvolution of BC (see section 2.3), on average $BC_{wb}$ represents 33 % and 29 % of total eBC in PM, at DEM and NOA, respectively. $BC_{ff}$ and $BC_{wb}$ fractions comprise the background concentration of BC ($BC_{bgd}$). Nevertheless, it was estimated that $BC_{bgd}$, defined as the 1.25 percentile of the dataset (Kondo et al., 2006), is below 10% of the arithmetic mean

concentration for both stations (Table 1). As a matter of fact, regional BC background concentrations are expected to be low compared to ambient levels in urban and suburban environments due to the low emission intensity of widespread sources. Wood burning contribution to total eBC is similar as in other European cities. Indeed, wintertime wood burning contribution of about 23-25 % has been reported for urban and suburban areas in Paris (Favez et al., 2009; Petit et al., 2014; Sciare et al., 2011), (24±11) % in Zurich downtown (Herich et al., 2011), and 23 % in London (Fuller et al., 2014).

Figure 5 and Fig. 6 present the diurnal cycle of $BC_{wb}$, $BC_{ff}$, $BC_{total}$ (=eBC) as well as the relative contribution of wood burning aerosols to the total BC (*WB%*). $BC_{wb,}$ as well as *WB%* show a clear diurnal trend, with values from 20-25 % early in the morning to peaks at 40 % during night-time, suggesting a large contribution of wood burning domestic sources spread over the region of Athens, in addition to the enhancement of concentrations at ground level because of the decrease in the boundary layer height. Other sources like industry and power generation are considered negligible as at European scales both consume

less than 1 % of the total amounts of wood used annually (Denier van der Gon et al., 2015; IEA, 2008). Fossil fuel source is nevertheless the main contributor to black carbon concentrations in both areas. In particular, during morning rush hours, it represents up to 70 % and 90 % of total eBC at DEM and NOA, respectively.

## 3.3 Source identification of CO

### 3.3.1 Using $BC_{wb}$ and $BC_{ff}$ as tracers of fossil fuel and wood burning sources

The association between CO concentration, $BC_{wb}$ and $BC_{ff}$ were examined using multiple linear regressions. Regression analysis between CO, $BC_{wb}$ and $BC_{ff}$ are shown for both sites in Fig. 7. Regressions were carried out using 10-minutes averaged data which represented a sample size of 13259 and 7474 values for DEM and NOA, respectively. The best-fitted linear equation to observed data, and the partial regression coefficients of the model $r'_0$, $r'_{ff}$ and $r'_{wb}$ were calculated so that:

$$[CO] = r'_0 + r'_{ff} \times [BC]_{ff} + r'_{wb} \times [BC]_{wb} , \qquad (20)$$

The model was run with no constraint for DEM, and the determined regression coefficients ($r'_{ff}$ and $r'_{wb}$ for DEM) were found with a precision (standard error of regression) below 2 %. However, for NOA, a constraint was applied in order to achieve a solution mathematically and physically meaningful. This choice was made because of the simultaneous advection of aerosols, resulting in a significant correlation between $BC_{wb}$ and $B_{ff}$ (Fig. 7), thus making more difficult the separation of different sources based on their variability. Since the variability of the emission ratios is greater for wood burning emissions (the emission ratio

strongly depends on type of biofuel and appliances used), the choice was made to constrain the emission ratio of fossil fuel $r'_{ff}$. The $r'_{ff}$ value for NOA was set identical to the one predicted by the model for DEM (i.e equal to 0.184 ppbv $ng^{-1}$ $m^3$, see Eq. 21). This is an approximation based on the assumption that BC and CO are chemically inert and their emission ratio $r'_{ff}$ cannot differ significantly within the same urban area due to an uniform vehicle fleet mix (vehicle size class and age of vehicle fleet, environmental performance, driving behaviour etc.). It should be noted here that this assumption might introduce some

uncertainty to the results. A sensitivity analysis for the NOA emission ratios was made, based on the statistical error of

determined $r'_{ff}$ at DEM, equal to one standard deviation in the regression analysis according to equation 20. Hence, $r'_{wb}$ for NOA was re-calculated using not only a single constraint value, but a range of values from the lowest (0.184-0.00137) to the highest (0.184+0.00137) around the determined $r'_{ff}$ value from the multilinear regression analysis at DEM. An uncertainty of 25% was finally estimated from this exercise for the calculated r'wb value at NOA. The results of the multiple regression analysis applied at the two sites are presented in Fig. 8.

$$CO_{ff-DEM}(ppbv) = (0.184 \pm 0.00137) \, x \, BC_{ff-DEM} \left(ng \, m^{-3}\right) \rightarrow (BC/CO)_{ff-DEM} = 5.4 \, ng \, m^{-3}/ppbv \,, \qquad (21)$$

$$CO_{wb-DEM}(ppb) = (0.114 \pm 0.00216) \, x \, BC_{wb-DEM}\left(ng \, m^{-3}\right) \rightarrow (BC/CO)_{wb-DEM} = 8.8 \, ng \, m^{-3}/ppbv, \qquad (22)$$

$$CO_{wb-NOA}(ppb) = (0.131 \pm 0.03275) \, x \, BC_{wb-NOA}\left(ng \, m^{-3}\right) \rightarrow (BC/CO)_{wb-NOA} = 7.6 \, ng \, m^{-3}/ppbv, \qquad (23)$$

The resulting regression coefficients were applied to estimate the fraction of CO attributed to fossil fuel and biomass burning combustion sources whereas the intercept values (108.5±0.64 and 146.8±2.5 ppbv and for DEM and NOA, respectively) were regarded as the background concentrations of CO. The resulting background concentrations are in very good agreement with those calculated as 1.25 percentile of the dataset (see Table 1). It is noteworthy to mention here that CO background levels are very significant with regard to the ambient concentrations, representing about 26% and 46% of the arithmetic mean concentration at NOA and DEM respectively. As a matter of fact, widespread natural sources of CO, such as plants, oceans and oxidation of hydrocarbons, in combination with its long atmospheric lifetime are known to maintain a significant background concentration even outside urban areas. These results show that the BC/CO ratio is higher for emission related to biomass burning compared to fossil fuel combustion, which is consistent with literature values (Pan et al., 2012). The determined values for DEM and NOA for wood burning are very similar, with a $(BC/CO)_{wb}$ ratio of 7.6-8.8 ng m$^{-3}$ ppbv$^{-1}$. These values are also in the low range of emission ratios found in the literature for both transport and domestic heating (using biofuel) sources (see Table 2).

The time series of the deconvolution of CO into three fractions, namely $CO_{ff}$, $CO_{wb}$ and $CO_{background}$, are shown in Fig. 9. According to our results, the wood-burning fraction of CO, represents on average 11 % and 16 % of total CO for DEM and NOA, respectively. In terms of concentrations, $CO_{wb}$ ranges between 5-52 ppbv ($\approx$ 25 ppbv on average) at DEM and between 2-406 ppbv ($\approx$ 135 ppbv on average) at NOA (Table 3). During night-time (20:00-02:00), this contribution is estimated at 15% and 25% respectively.

Diurnal variabilities of $CO_{ff}$ and $CO_{wb}$ at each site are presented in Fig. 10 (*NB*: different vertical scales are used for each station). As expected, $CO_{ff}$ presents similar variability with that of $BC_{ff}$, i.e a pronounced bimodal distribution, with higher concentrations during rush hours. Along the same line, the diurnal variability of $CO_{wb}$ shows a unimodal pattern with increasing concentrations after 18:00, due to the combination of enhanced wood burning emission which is a source more active during evening hours and lower ambient temperature and BLH, as discussed previously.

Comparison with other European cities is limited by the very few number of studies investigating the carbon monoxide concentrations sources. Saurer et al. (2009) used the stable isotope composition of CO ($\delta^{13}$C and $\delta^{18}$O) for the characterization of different CO sources at 3 sites in Switzerland during winter (along with other indicators for traffic and wood combustion such as NO$_x$-concentration and aerosol light absorption at different wavelengths) and estimated the wood burning contribution to night-time CO concentrations at 70%, 49% and 29% for a village site dominated by domestic heating, a site close to a motorway and a rural site, respectively. These differences reflect the spatial variability in the wood burning use within the same region depending on the type of site, as well as between countries depending on the regional heating practices.

### 3.3.2 Comparison the CO-BC$_{wb}$-BC$_{ff}$ linear model vs the CO-NO$_x$ linear model

The results of the multiple-regression model were compared with those from the CO-NO$_x$ linear model. As presented in section 2, an *a priori* knowledge of the emission ratio of CO/NO$_x$ from traffic ($r_{ff}$) and wood burning ($r_{wb}$) emissions is required for the CO-NO$_x$ model and therefore the choice of their value is of major importance.

In Fig. 11, the scatter plot of CO versus $NO_x$ is shown for the data recorded at DEM and NOA. The data set is mostly comprised between two well defined slopes and the range of values of $CO/NO_x$ observed in actual air samples is 7-25 ppbv ppbv$^{-1}$ (after subtraction of the $CO_{bgd}$ concentration) for DEM and NOA stations and depends on the contribution of each of the sources. As applied in other studies for similar purposes (Rodríguez and Cuevas, 2007; Saurer et al., 2009), these slopes were estimated using two best fit lines, the first to the points aligned in the lower edge of CO versus $NO_x$ scatter charts, and the second one to the points aligned in the upper edge respectively. More precisely, they have been calculated by fitting only data below the 10th percentile and above the 90th percentile respectively of $CO/NO_x$ data. Knowing that the lowest ratios are obtained when traffic emissions dominate (and when the contribution of woodburning is insignificant), an $r_{ff}$ value of 7 ppbv ppbv$^{-1}$ was estimated. Higher ratios were obtained typically late in the night-time, when traffic emission decreased and domestic heating increased. However, as we do not expect a contribution of domestic heating close to 100 % at any time of the day in Athens, it is impossible to estimate with accuracy the $r_{wb}$ ratio based solely on this data. Nevertheless, based on "wood burning" lines from Fig. 11, and assuming that emission ratios from wood burning are similar between NOA and DEM, we estimate a $r_{wb}$ ratio for the area of Athens, larger than 25 ppbv ppbv$^{-1}$.

A sensitivity analysis of the CO-$NO_x$ model was performed using the experimentally determined $r_{ff}$ (at 7 ppbv ppbv$^{-1}$) and by varying $r_{wb}$. Measurements performed directly at the emission source and close to the chimney exhausts during controlled wood-burning experiments, indicated ratios in the range of 50-150 ppbv ppbv$^{-1}$(Albinet et al., 2015; Nalin, 2014). As a result, the choice was made to vary the $r_{wb}$ parameter from 50 to 150 ppbv ppbv$^{-1}$. The results of the analysis are presented in Table 4. The influence of increasing $r_{wb}$ at a constant $r_{ff}$ resulted in a relatively minor reduction in the calculated contribution from wood-burning of 2-3 %. A good correlation was found between both models, with coefficients of determination $R^2$ of 0.52 and 0.85 at DEM and NOA, respectively. The wood burning contribution to CO estimated using the CO-$NO_x$ linear model is higher compared to the estimations from CO-$BC_{ff}$-$BC_{wb}$ model, by a factor of about 1.5 for NOA, which can be considered as "acceptable" given the uncertainties associated with both models. A higher overestimation compared to model 2 by a factor of about 2.4 is found at DEM suburban station, which could be explained by the fact that the site is characterized by more aged air masses compared to the urban background NOA station. Consequently, ambient $CO/NO_x$ ratios might differ more significantly from emission ratios at DEM suburban station. These results suggest that the CO-$NO_x$ linear model probably overestimate wood burning contribution to CO, especially in environments characterized by aged air masses were photochemical loss of NOx cannot be considered as negligible.

## 4. Conclusion

In this study we performed a comprehensive field campaign at two surface stations in a suburban and a central area of Athens during winter 2014-2015 in order to investigate the impact of fossil fuel and biomass combustion on the urban air quality. We report measurements of particulate black carbon and CO and $NO_x$ gaseous components performed simultaneously at the monitoring station of Demokritos (DEM), representative of suburban areas of the Athens Metropolitan Area and at the National Observatory of Athens (NOA), typical of urban background conditions. More precisely, black carbon particles were concurrently measured using two 7-wavelength aethalometers, whereas mixing ratios of CO were measured with an infrared absorption analyzer at NOA and by wavelength-scanned cavity ring down spectrometry at DEM.

The median BC concentrations were 528 ng m$^{-3}$ at DEM and 1198 ng m$^{-3}$ at NOA. In a similar way, median CO mixing ratios were 195 ppbv at DEM and 324 ppbv at NOA. These differences have been explained by the location of the two sites with respect to the proximity from sources and local atmospheric dynamics in the Athens valley. Both BC and CO displayed a clear bimodal diurnal pattern, in which morning peaks were observed due to morning inversion and rush-hour traffic, while evening peaks were attributed to combustion sources (evening traffic rush-hour, residential heating) combined with the effects of a

shallow nocturnal boundary layer. The highest concentrations were observed during low wind speeds, suggesting that both combustion products were not related to regional transport but instead originated from sources within Athens.

Source apportionment of BC was carried out using a model based on the absorbance spectral differences of black carbon (related to fossil fuel) and brown carbon (related to biomass burning). Our results suggest that even though fossil fuel combustion is the major contributor to BC in $PM_{10}$, woodburning makes an important contribution of about 30% to wintertime BC concentrations at both sites (on average 33 % at DEM and 29 % at NOA, respectively, but this difference lies within the uncertainty range of the calculations). BC from biomass burning displayed a clear unimodal diurnal pattern with the highest concentrations during night, confirming that its main source was local domestic heating.

As both datasets showed significant BC and CO correlations, we used observations of CO mixing ratios along with the fraction of $BC_{wb}$ and $BC_{ff}$ to quantify the percentage of observed CO which originates from fossil fuel and wood burning sources. This analysis led to the conclusion that the wood-burning fraction of CO from local emissions, represents on average 11-16 % of total CO in Athens during wintertime. The method proposed here for the source apportionment of CO was compared to a previously reported method based on the $CO$-$NO_x$ ratios. From our results, it appears that the $CO$-$NO_x$ linear model over-estimates the contribution of wood burning to CO concentrations, especially in environments characterized by aged air masses, likely due to the fact that the hypothesis of negligible photochemical loss of $NO_x$ is not always met.

**Acknowledgements**

Financial support from the European Union's EnTeC FP7 Capacities programme (REGPOT-2012-2013-1, FP7, ID: 316173) and the Horizon 2020 research and innovation programmes under grant agreements No 654109 and No 689443 is kindly acknowledged. Thanks are due to prof. Jean Sciare for the loan of the AE-42 aethalometer at Thission. We also acknowledge financial support by the KRIPIS/NSRF2007-2013 project and thank NOA team (Drs V. Psilloglou and M. Lianou) for the operation and maintenance of NOA's site at Thission. This study contributes to ChArMEx work package 1 on Emissions and Sources.

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

**Table 1. Statistical summary of eBC and CO concentrations at DEM and NOA stations for the period between 6th**
**December 2014 and 10th March 2015.**

| | Arithmetic mean ± stdv | Geometric mean | Median | 90th percentile | 10th percentile | 1.25th percentile[**] | Number of datapoints (10-min averaged) |
|---|---|---|---|---|---|---|---|
| eBC (DEM) $ng.m^{-3}$ | 656±519 | 497 | 528 | 1265 | 179 | 71 | 13259 |
| eBC (NOA) $ng.m^{-3}$ | 2655±3554 | 1372 | 1198 | 6963 | 352 | 57 | 7474 |
| CO (DEM) $ppbv$ | 214±95 | 199 | 195 | 315 | 125 | 101 | 13259 |
| CO (NOA) $ppbv$ | 555±570 | 404 | 324 | 1282 | 196 | 143 | 12127 |
| $NO_x$(DEM) [*] | 6.17±3.47 | | 5.56 | 9.93 | 3.66 | 3.13 | 2238 |
| $NO_x$ (NOA) | 29.6±42.7 | 12.4 | 11.4 | 84.5 | 1.9 | 1.2 | 12127 |

[*]$NO_x$ measurements were available from the monitoring station of the Greek Ministry of Environment and Energy (www.ypeka.gr) situated 300 m from DEM station

[**]Background concentration levels estimated as in Kondo et al., 2006

**Table 2. BC/CO (ngC m$^{-3}$ ppbv$^{-1}$) ratios derived from emission factors found in the literature and from ambient measurements in Athens using the multiple regression model.**

| | Transport | Domestic heating (biofuel) | Reference |
|---|---|---|---|
| *Derived from Emission factors* | | | |
| | 3.1-11.5 (gasoline) 1.3-55 (diesel) | 8.7-26.6 | Verma et al., 2010 |
| *Derived from ambient measurements (multiple regression model)* | | | |
| DEM station | 5.4 | 8.8 | This study |
| NOA station | 5.4 (fixed) | 7.6 | This study |

**Table 3. Statistical summary of calculated CO$_{wb}$ and CO$_{ff}$ concentrations, as well as CO$_{wb}$(%) using the multiple regression model (model 1) at DEM and NOA stations.**

| | DEM | | | NOA | | |
|---|---|---|---|---|---|---|
| | $CO_{wb}$ (ppbv) | $CO_{ff}$ (ppbv) | $CO_{wb}$ (%) | $CO_{wb}$ (ppbv) | $CO_{ff}$ (ppbv) | $CO_{wb}$ (%) |
| mean (±stdv) | 25.1 (25.7) | 80.2 (65.5) | 11 (9) | 134.5 (230.6) | 297.3 (366.9) | 16.3 (14.3) |

| | | | | | |
|---|---|---|---|---|---|
| 90th percentile | 51.9 | 156.7 | 21 | 405.9 | 731.0 | 37.1 |
| 10th percentile | 4.6 | 19.8 | 3.0 | 1.9 | 50.8 | 1.0 |

**Table 4. Sensitivity test of the CO-NO$_x$ linear model for a constant emission ratio for traffic ($r_t$) and a variable emission ratio for wood burning and comparison of model 1 (linear) versus model 2 (multilinear).**

| | $r_t$ (ppbv/ppbv) | $r_{wb}$ (ppbv/ppbv) | Mean $CO_{wb}$/CO (%) | Slope of regression between $(CO_{wb})_{model1}$ and $(CO_{wb})_{model2}$ |
|---|---|---|---|---|
| DEM station | | | | |
| Test 1 | *7* | *50* | 29% | 2.4 (R$^2$=0.52) |
| Test 2 | *7* | *100* | 28% | 2.2 (R$^2$=0.52) |
| Test 3 | *7* | *150* | 26% | 2.1 (R$^2$=0.52) |
| NOA station | | | | |
| Test 1 | 7 | 50 | 34% | 1.6 (R$^2$=0.85) |
| Test 2 | 7 | 100 | 31% | 1.51 (R$^2$=0. 85) |
| Test 3 | 7 | 150 | 30% | 1.48 (R$^2$=0. 85) |

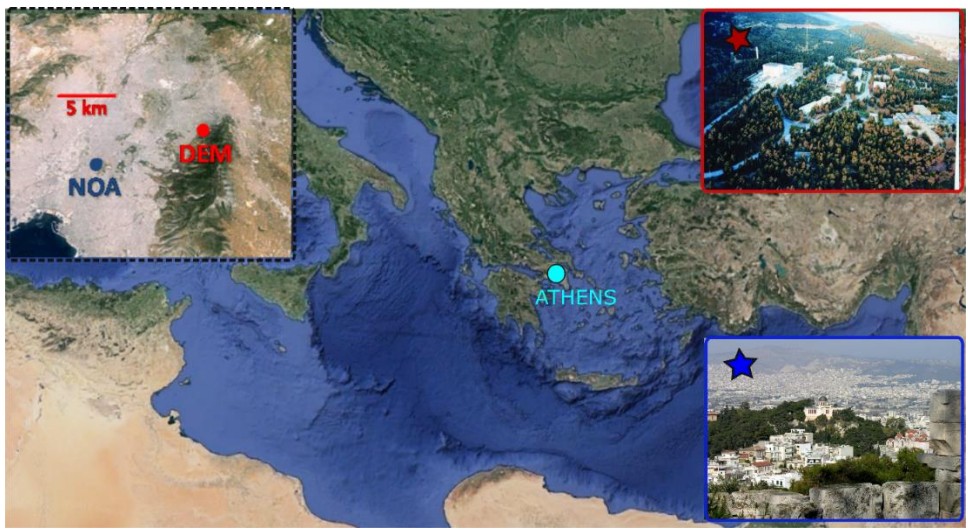

Figure 1: Regional map along with 3D satellite map of the Athens Metropolitan area (black dashed rectangle), and photos of the NCSR Demokritos (DEM) campus in Agia Paraskevi (Athens suburban, red star), and of the National Observatory of Athens (NOA) at Thiseion (Athens centre, blue star).

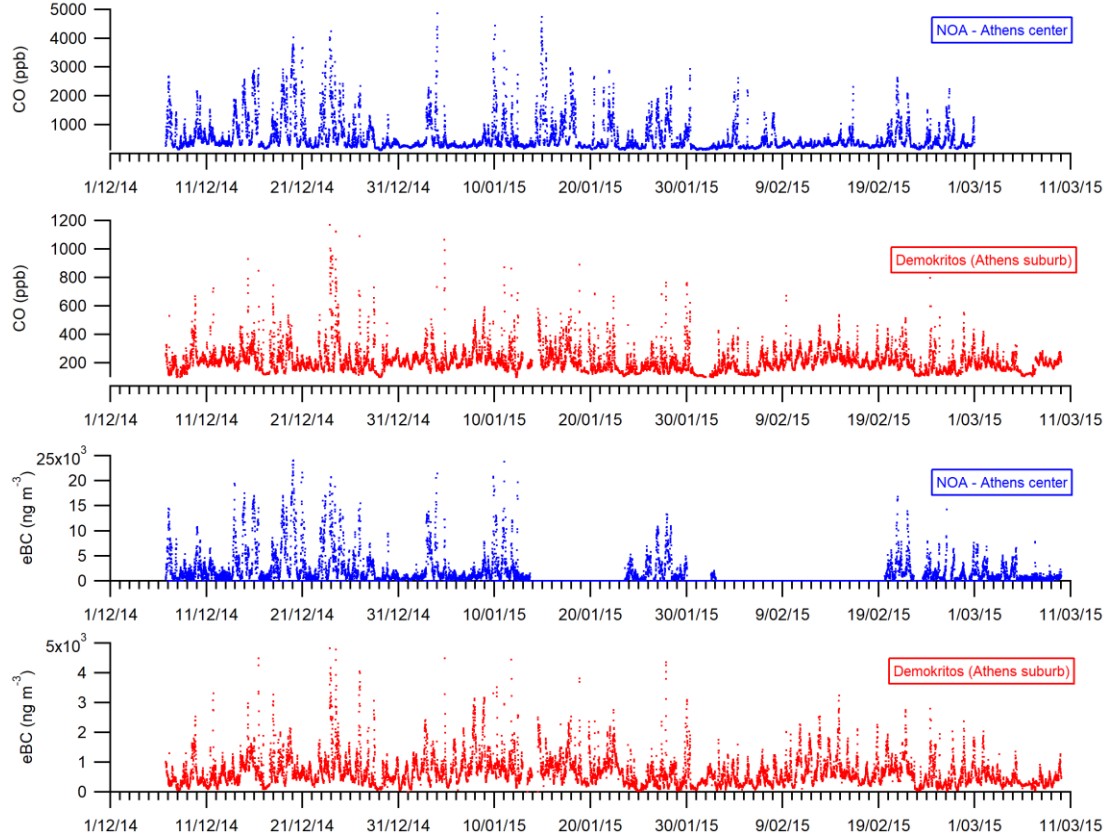

**Figure 2: Time series of 10-minutes averaged CO and eBC concentrations measured at NOA (in blue) and DEM (red) monitoring stations from 6th December 2014 until 10th march 2015.**

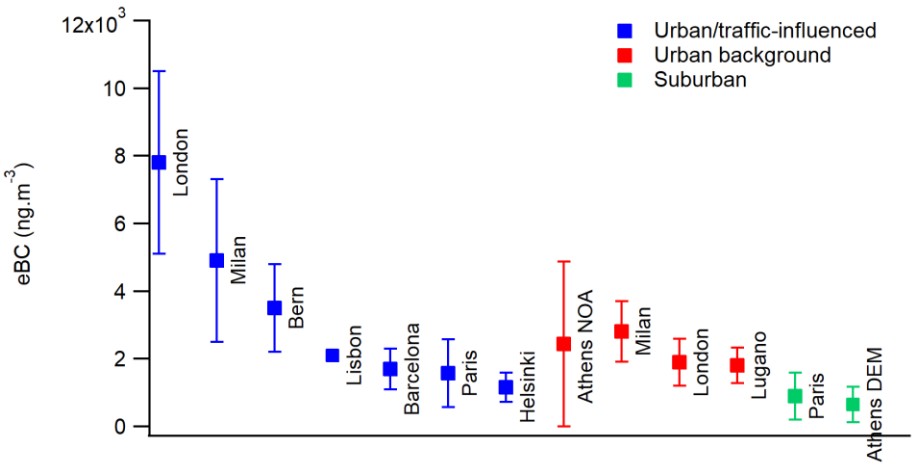

**Figure 3: Levels of black carbon (mean±stdv) reported for various European cities, (blue) in traffic-influenced sites, such as London (year 2009, Reche et al., 2011), Milan (Summer 2010, Lisbon (June 2000, Alves et al., 2002), Barcelona (year 2009, Reche et al., 2011), Paris (summer 2009, Zhang et al., 2013) and Helsinki (winter 2000, Hitzenberger and Tohno, 2001); (red) in urban background/residential areas such as Milan (Summer 2010, Invernizzi et al., 2011), London (year 2009, Reche et al., 2011) and Lugano**
10 **(year 2009, Reche et al., 2011), (green) a suburban area of Paris (Laborde et al., 2013), along with the results from this study at NOA and DEM stations in winter 2014-2015.**

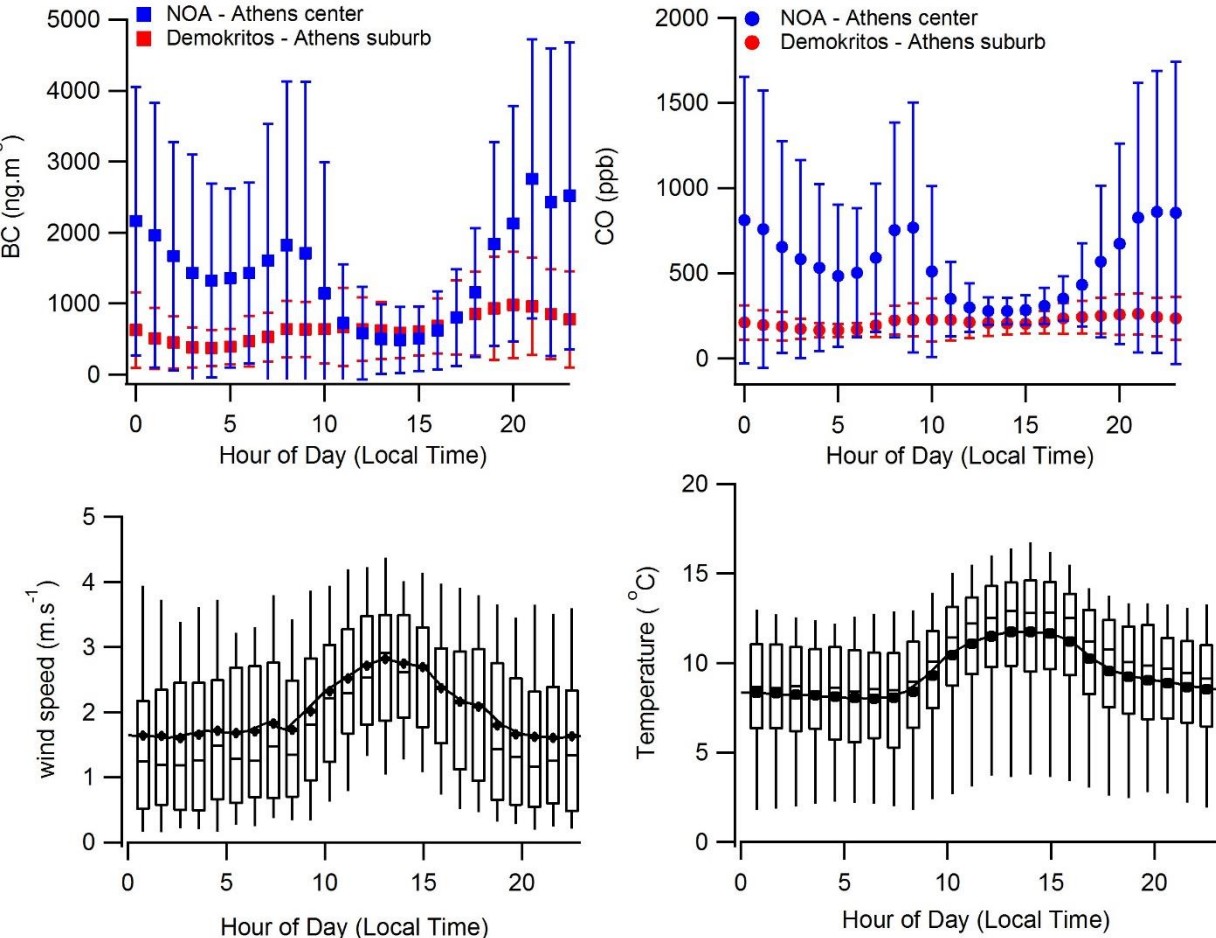

**Figure 4: (Top) Diurnal trends of BC (left) and CO (right) concentrations at NOA (blue) and DEM (red) monitoring stations (Local Time = UTC). Vertical bars show standard deviation to the mean value. (Bottom) Diurnal trends of wind speed (left) and temperature (right) measured at the meteorological station of Demokritos. Data are presented as box and whisker plots, where boxes encompass values between the 25th and 75th percentiles, horizontal lines represent median values, and 'whiskers' give the 80% range of the values, whereas markers represent the mean values.**

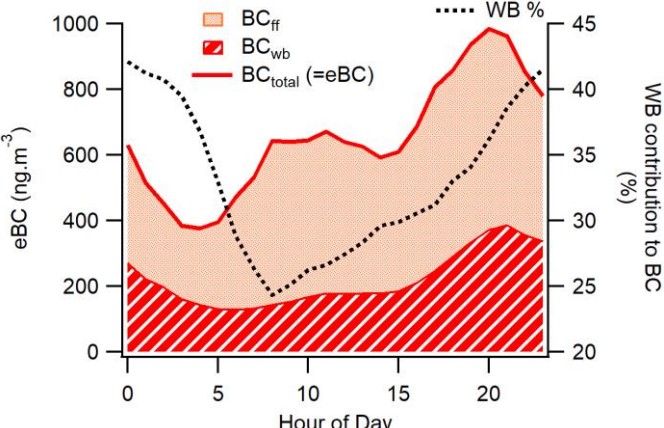

**Figure 5: Diurnal cycle of $BC_{wb}$, $BC_{ff}$, $BC_{total}$ and wood burning contribution to total BC (WB, defined as $BC_{wb}/ BC_{total}*100$) at DEM station.**

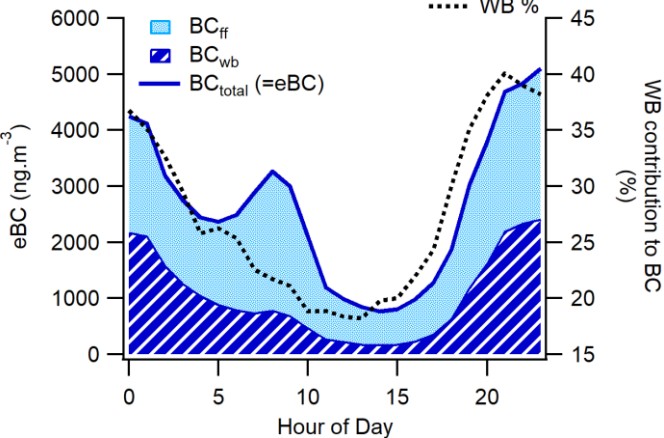

**Figure 6: Diurnal cycle of $BC_{wb}$, $BC_{ff}$, $BC_{total}$ and wood burning contribution to total BC (WB, defined as $BC_{wb}/ BC_{total}*100$) at NOA station.**

**Figure 7: Correlation plots between (left) CO and $BC_{ff}$, (middle) CO and $BC_{wb}$, (right) $BC_{wb}$ and $BC_{ff}$ for DEM (a) and NOA (b) stations.**

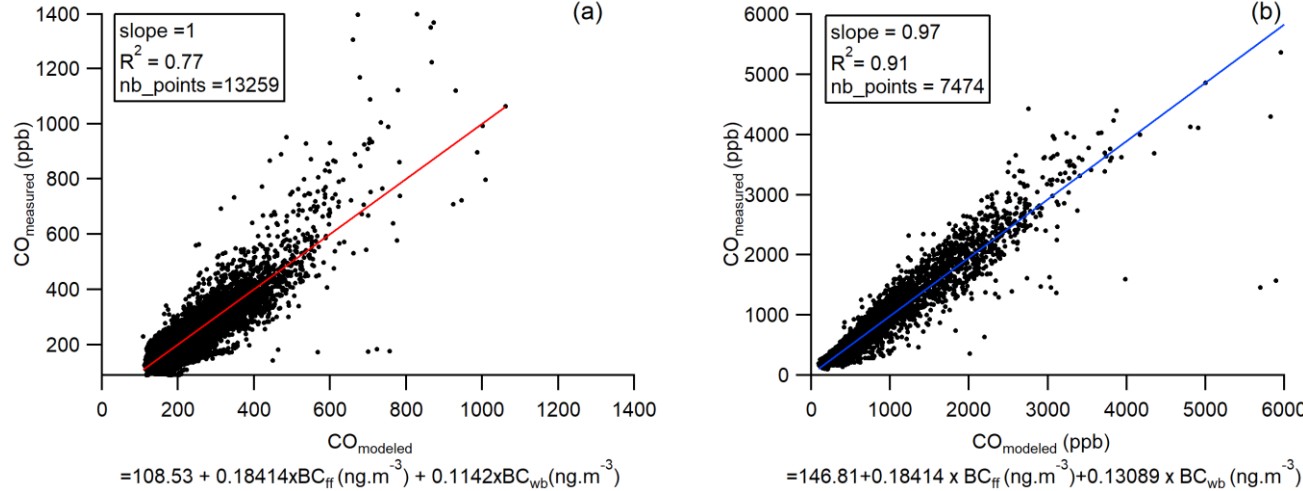

**Figure 8: Best-fit linear correlations between CO and a combination of $BC_{ff}$ and $BC_{wb}$ for DEM (a) and NOA station (b).**

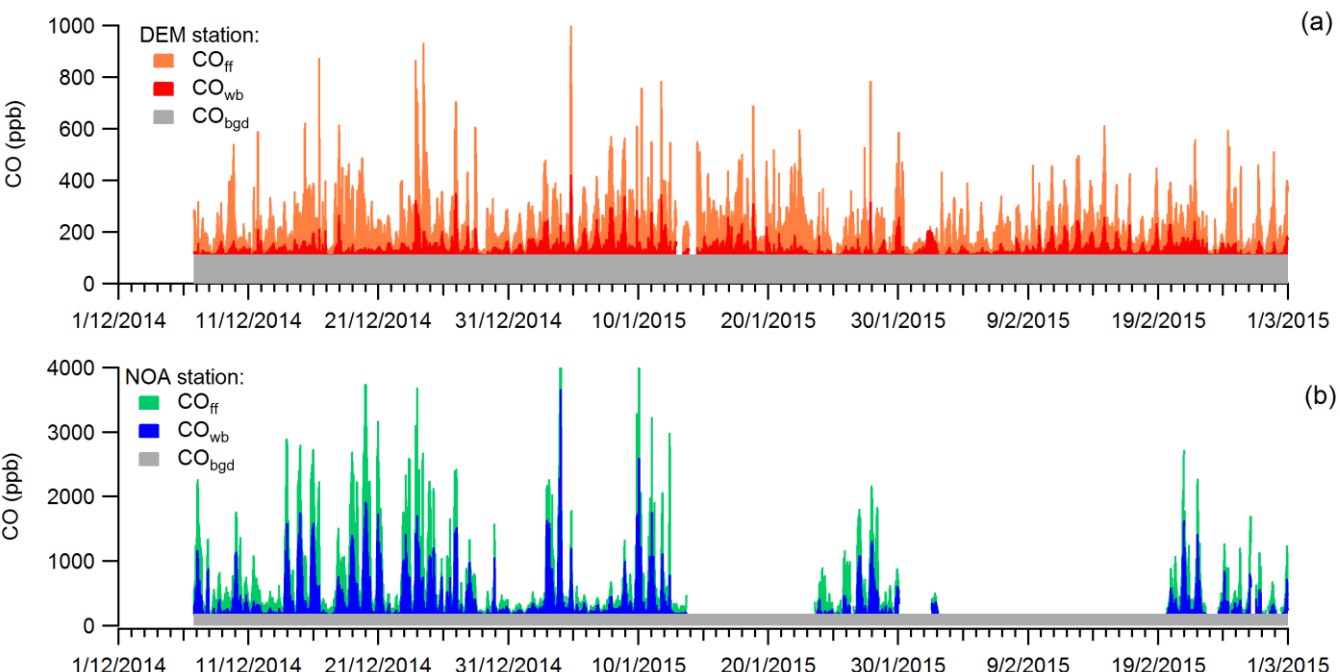

5     **Figure 9: Time series of the calculated $CO_{ff}$, $CO_{wb}$ and $CO_{bgd}$ concentration at (a) DEM and (b) NOA stations.**

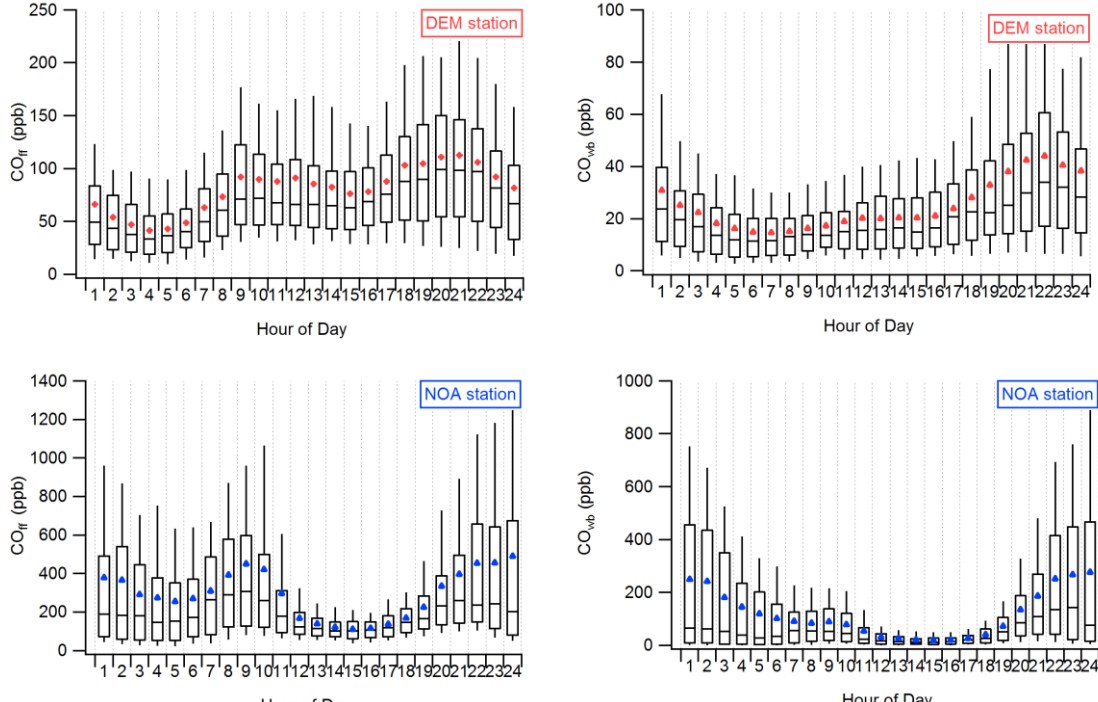

**Figure 10: Diurnal variations of CO$_{ff}$ (left) and CO$_{wb}$ (right) at DEM (top) and NOA (bottom) stations.** Data are presented as box and whisker plots, where boxes encompass values between the 25th and 75th percentiles, horizontal lines represent median values, and 'whiskers' give the 80% range of the values, whereas colored markers represent the mean values.

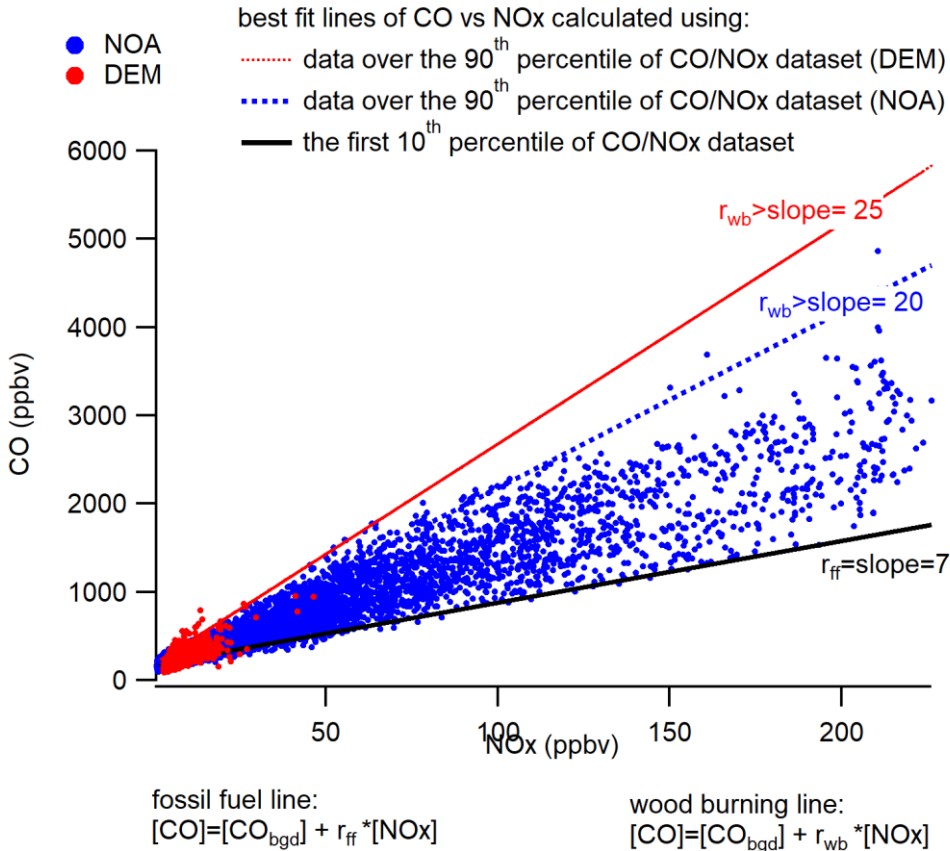

**Figure 11: Scatter plot for CO and NO$_x$ data from NOA (blue) and DEM (red) along with best fit lines, aligned in the lower edge of CO versus NO$_x$ scatter charts (black), and in the upper edge (blue and red for NOA and DEM, respectively).**

