# Peer review of "Assessment of wood burning versus fossil fuel contribution to wintertime black carbon and carbon monoxide concentrations in Athens, Greece"

_Atmospheric Chemistry and Physics, 2017_

## Referee Comment (RC1) · Anonymous Referee #1 · 19 Dec 2017

This manuscript addresses up-to-date scientific questions within the scope of the journal, and may indeed be considered as relevant for the special issue dedicated to ChArMEx. Its overall presentation (including the title, the abstract and the figures) is appropriate, clear and globally well structured.

It presents results of wintertime BC and CO source apportionment results obtained for Athens, Greece. To do so, authors claim they are using two different and independent methodologies. However, the "CO/NOx ratio" approach appears to be irrelevant in the present case, so that outputs are not used for the purpose of the study. My only major

comment is related to this latter issue, and I would recommend presenting the use of the "CO/NOx ratio" approach in another way (or to simply skip it).

More specific comments are listed below: - Abstract, line 21: occasional significant impacts of long-range transport are not really discussed/demonstrated in the manuscript. - Page 2, lines 16-23: it is not clear within which periods the discussed increases/decreases were observed (e.g., lines 16-18: a constant increase of 30% every year since 2012 ? or an increase of 30% for the period 2012-201x, compared to which period ? ...). - Page 3, line 9: how much "relatively far" from major roads? - Page 3, lines 18-26: it may be worthy to indicate more clearly this data correction procedure was applied to AE42 (and AE31) datasets only. Also, what could be the uncertainties related to the use of $f\lambda$ values that weren't estimated for this individual instrument / site? Finally, what could be the impact of the PM10 cut-off, compared to the PM2.5 used at the other site ? - Page 4, line 6-7: please indicate whether this value was also obtained using the 1.64 "ACTRIS correction factor" (as used by Zanatta et al., 2016) ? - Page 4-5, BC and CO source apportionment: please discuss here possible interference from coal combustion emissions. - Page 5, BC source apportionment: please justify/discuss a bit more the choice of $\alpha$wb = 2 by comparison with value recently proposed by Zotter et al., 2017. - Page 6, line 36 - Page 7, line 7: the expected diurnal cycle of the intensity of emissions could be discussed more deeply here. - Page 8, line 25 (% COwb): please discuss these percentage regarding previous studies/results. - Page 9, line 25-26: here, it sounds like wind speed is controlling the diurnal patterns. Please consider rephrasing this sentence. - Figure axes: please homogenize the use of "BC" / "eBC". - Figure 7, right panel: legend of the y-axis seems inaccurate.
* * *

---

## Referee Comment (RC2) · Anonymous Referee #1 · 21 Dec 2017

As asked by Associate Editor, please find herewith clarifications about RC1 referee major comment related to the use of the "CO/NOx ratio" approach.

From what authors are writing in the abstract and introduction, it sounds like they are actually using results they obtained from this approach: - Abstract, lines 19-20: "For an independent evaluation of the results, we additionally estimated the wood-burning and fossil fuel contribution to CO, calculated on the basis of their CO/NOx emission ratios" - P.2, lines 41-42: "Two independent methods based on the relations between CO and co-products of combustion processes were used and compared for the esti-

mation of CO originated from traffic and wood combustion." However, when presenting/mentioning results from the "CO/NOx ratio" approach, authors do not provide results from this approach about CO source apportionment (i.e., are not using these outputs to estimate the contribution of biomass burning emissions to CO). Instead, the reader is learning there that "CO-NOx linear model always over-estimated the wood burning contribution to CO, compared to the CO-BCff-BCwb model. The overestimation of the CO-NOx linear model is probably the result of daytime photochemical loss of NOx that is not taken into account in our study." (P.9, lines 12-14). I would then assume that this approach is eventually not accurate in the present case, and is therefore not used here to estimate the wood-burning and fossil fuel contribution to CO. For this reason, I would strongly recommend presenting the use of the "CO/NOx ratio" approach in another way (or to simply skip it).
* * *

---

## Referee Comment (RC3) · Anonymous Referee #2 · 22 Dec 2017

General The manuscript attributes the concentration-contribution of wood burning to air pollution in Athens and with that handles an important health related subject that needs attention. For the source apportionment of black carbon, the well-known technique based on wavelength dependence of the aerosol light absorption coefficient is used after application of necessary compensations. For CO apportionment two models are described. Model 1 relies on known emission ratios of NOx and CO and assumed similar atmospheric lifetimes. This model is not trusted by the authors and according to the authors this linear model always over-estimates the wood burning contribution.

[Figure]

The results from model 1 are not researched enough to give recommendations to other scientific studies. It remains unclear why this model is included in this paper. A short discussion in the introduction could be enough. Model 2 for CO apportionment is based on multiple linear regression between CO and BC. This technique is well-known but in the conclusion section it is written that the method is new; or is the application to CO in combination to wood burning new? If new this fact should receive more attention earlier in the paper.

The manuscript is not pushing scientific boundaries, but contains important numbers, e.g. 30% of BC is wood burning related.

The authors were a bit sloppy on the correct references.

Details Abstract: '30% to the observed eBC and the CO concentrations (….)' this doesn't read well and is confusing for CO.

Section 2.2.1 For compensation parameter f values given in Drinovec et al. (2015) were used. But Drinovec et al. describe that filter loading effects change with location and time. The values of Drinovec differ a lot from Sandradewi et al (2008) (reference in manuscript but not listed in References, make sure to find the correct paper) and Zotter et al (2017). The latter paper, that is known to the authors confirms the importance of proper compensation. I would like to see a more worked out compensation correction.

About the instruments At DEM (AE33) at NOA(AE42). I understand that AE33 and AE31 are compared. At page 3 line29: 'and data from AE31 aethalometer with …….to AE-42, which operates continuously in parallel with the AE33 at DEM station. So what instruments are compared and where is the AE-31 located?

The R-squared of 0.79 is not very convincing for aethalometers, I would like to see the plot. The intercept (what is the unit? Inverse Mm or ng/m3?) of 0.15 is interesting.

I'm not sure, if it is interesting that 'BC is historically defined from aethalometer measurements at 880nm'. The important message should be that the whole spectral dependence approach depends on fixation somewhere. This is done at 880nm because it is believed that at that wavelength the MAC for wood burning and fossil fuel combustion is very close. Otherwise the DEC MAC cannot be applied at NOA. The whole fractitioning is based on the wavelength dependence that is somewhere fixed (Equation 10). The reader should be convinced of the choice that is supported by literature.

It is written '(MAC). . .. (determined from the comparison with concurrent measurements at DEM of elemental carbon)'. A bit later a reference to Diapouli et al. (2014) is included. Does this paper include the 7.5 m2 g-1?

The angstrom exponent for absorption is measured why do the authors assume an exponent of 1.0 in line 6 (p4).?

P4 line21 Reference to Sandradewi (please include the correct one in references). Sandradewi discussed different Angstrom exponents depending on the chosen wavelengths. This wavelength dependence should be discussed in light of the choices given in line 32, or refer to other studies that use same wavelengths. The 470 nm channel was broken in that Sandradewi study, why does this study start at 470 nm (line 32)

P5 top para. Exponents 0.9 (traffic) and 2.0 (wood) 'were used, based on the range of values.. reported'. The value of 2.0 is disqualified by Zotter et al., 2017, because is leads to differences with radiocarbon results. The exponents are crucial to the method, 'based on' should be worked out.

P6 line 20 'the hypothesis of negligible photochemical chemistry is validated.' Where is it validated please include reference(s). Negligible what does that mean, negligible for the scale considered in this study? Or is the assumption that BC and CO have similar lifetimes? This para needs to be worked out to convince the reader that model2 is superior to model 1.

Page 7 line 9 'eBC in PM2.5' but at NOA a PM10 sampling head is installed, right?

Page7 line 11. Apparently the 1.25 percentile of a dataset can be used for background.

Really would like to read that paper. Please include Kondo et al., 2006 in the references.

P7L14 'relatively short lifetime of BC' please compare to P6L20 P7L34 how is the relative standard deviation defined in this case. P7L39 '0.184' please include units if appropriate

P8L4 the value 0.00137 is 0.7% of the best estimate 0.184. This is very small compared to values in Table 2. Please include discussion. Why is 0.00137 an useable value and why is the resulting uncertainty of 25% for the emission ratio 'rather reasonable' (not scientific terminology)

P8L13 background or intercept values are 109 and 147 how do these differences related to 'cannot differ significantly' line 1 of this page?

P8L14 background concentrations of CO with a reference to Goldstein and Schade (2000), this work contains some informations on background but not on CO. How should the reader interpret the reference, please modify.

P8L14etc The resulting background concentration are in very good agreement with . . . 1.25 percentile. Really want to learn more. For me it sounds like abracadabra.

P8L41 'using a best fit line' If this is a fit how was the data selected? This was not clear from the references literature.

P9Line4 informs us that the ratio is larger than . . .please explain

P9L9 'values found in the literature' please include references.

P9L10 2-3% where should I look to see the supporting material.

Table 4 Regression Slope between model 1 and model 2: what model outcomes are regressed? Are we looking at COwb/totalCO?

Typos-suggestions P3 line26 'this purpose' → for loading compensation P4 L9 ratios

were P4 line 25 lambda is bold in equation P4 eq 5 lambda1 should be lambda2 in denominator. P7L40 last ff should be sub. P8L28 diurnal variabilities . . .are P8L33 Comparison of A and B Figure 7 caption or axis titles are wrong for right bottom figure

---

## Author Response (AR1)

**Assessment of wood burning versus fossil fuel contribution to wintertime black carbon and carbon monoxide concentrations in Athens, Greece**

**Authors Responses to referees' comments**

The authors acknowledge both referees for critically reading the manuscript and for their contribution in improving and clarifying this study. The authors have compiled the responses as follows. Reviews from Referee 1 and Referee 2 are in blue and Brown font respectively, and have been grouped based on the section of the manuscript they refer to. Author responses are in black font numbered with [A0, A1, A2 …]. Italics and quotations are used for the information added in the revised manuscript.

**General Comments**

- Reviewer 1: This manuscript addresses up-to-date scientific questions within the scope of the journal, and may indeed be considered as relevant for the special issue dedicated to ChArMEx. Its overall presentation (including the title, the abstract and the figures) is appropriate, clear and globally well structured. It presents results of wintertime BC and CO source apportionment results obtained for Athens, Greece. To do so, authors claim they are using two different and independent methodologies. However, the "CO/NOx ratio" approach appears to be irrelevant in the present case, so that outputs are not used for the purpose of the study. My only major comment is related to this latter issue, and I would recommend presenting the use of the "CO/NOx ratio" approach in another way (or to simply skip it).

- Referee 2: The manuscript attributes the concentration-contribution of wood burning to air pollution in Athens and with that handles an important health related subject that needs attention. For the source apportionment of black carbon, the well-known technique based on wavelength dependence of the aerosol light absorption coefficient is used after application of necessary compensations. For CO apportionment two models are described. Model 1 relies on known emission ratios of NOx and CO and assumed similar atmospheric lifetimes. This model is not trusted by the authors and according to the authors this linear model always over-estimates the wood burning contribution. The results from model 1 are not researched enough to give recommendations to other scientific studies. It remains unclear why this model is included in this paper. A short discussion in the introduction could be enough.

- A0: Both referees major comment concerns the use and presentation of the CO-NOx linear model used for CO source apportionment. From their comments, we understand that the reason why the model was used seemed unclear since at the end, we give more confidence to the second model.

    However, we believe that since the CO/NOx ratio has been used in the past as a diagnostic ratio to characterize different type of emission sources (Fujita et al., 1992; Ravindra et al., 2006; Wahlina et al., 2001), it is interesting to discuss the output of the model 1 with respect to the output of the model 2. Additionally, this model could be of use in the case where no absorption measurements are available. This could be for instance the case at monitoring stations part of national networks where only regulated air pollutants are often measured. In order to make a clearer introduction to the reason why we have selected to use this model, we have added in the manuscript, section 2.4.1, the following:

    o "P5L29-32: "*The CO/NOx ratio has been used in the past as a diagnostic to characterize different type of emission sources* (Fujita et al., 1992; Ravindra et al., 2006; Saurer et al., 2009; Wahlina et al., 2001)*. It can serve as a useful tool for apportioning CO concentrations at monitoring stations part of national networks where only regulated air pollutants are often measured.*"

    We have also added some discussion about the conditions limiting the use of the model. Indeed, our results suggest that CO-NOx should be used with more caution in environments dominated by aged air masses. As a matter of fact, at the NOA urban background site, the difference between both models is of a factor of about 1.5, which can be considered rather acceptable given the level of uncertainty associated with source apportionment methods. Higher difference between both models are observed for the DEM suburban site. Finally, since this type of model is being used in other studies, we suggest that it is a benefit for future studies to keep the results of this comparative assessment against model 2 in order to have documented the results of the current evaluation.
    The following discussion was added in the manuscript
    o P10L01-08: "*The wood burning contribution to CO estimated using the CO-NO$_x$ linear model is higher compared to the estimations from CO-BC$_{ff}$-BC$_{wb}$ model, of a factor of about 1.5 for NOA, which can be considered as "acceptable" given the uncertainties associated with both models. A higher overestimation compared to model 2 of a factor of about 2.4 is found at DEM suburban station, which could be explained by the fact that the site is characterized by more aged air masses compared to the urban background NOA station. Consequently, ambient CO/NO$_x$ ratios might differ more significantly from emission ratios at DEM suburban station. These results suggest that CO-NO$_X$ linear model probably overestimate wood burning contribution to CO, especially in environments characterized by aged air masses were photochemical loss of NOx cannot be considered as negligible.*

- Model 2 for CO apportionment is based on multiple linear regression between CO and BC. This technique is well-known but in the conclusion section it is written that the method is new; or is the application to CO in combination to wood burning new? If new this fact should receive more attention earlier in the paper. The manuscript is not pushing scientific boundaries, but contains important numbers, e.g. 30% of BC is wood burning related.
    o We have taken into account the referee comments and added the following in the revised manuscript, in order to address what is the novelty about the method used.

      ○  P6L38-40: *"While multiple linear regressions are known techniques for source apportionment, they have not yet been applied to investigate wood burning contribution to CO using the aethalometer's model results. The considerable increase in measurements carried out using aethalometers makes this technique an interesting and very useful methodology for apportioning CO concentrations."*

**Specific comments**

**Abstract**

- line 21: occasional significant impacts of long-range transport are not really discussed/demonstrated in the manuscript.
  - A1: The sentence "*and are only occasionally affected by long-range transpo*rt" has been removed from the manuscript.

- 30% to the observed eBC and the CO concentrations (: : :.)' this doesn't read well and is confusing for CO.
  - A2: Corrected in the revised manuscript (P1L23).

**1 Introduction**

- Page 2, lines 16-23: it is not clear within which periods the discussed increases/decreases were observed (e.g., lines 16-18: a constant increase of 30% every year since 2012 ? or an increase of 30% for the period 2012-201x, compared to which period ? : : :).
  - A3: Precisions on the time periods discussed have added in the revised manuscript: P2L16,L18,L20

**2 Material and Methods**
**2.1 Sampling Sites**

- Page 3, line 9: how much "relatively far" from major roads?
  - A4: The station is at about 500 m from major roads (this information was added in the revised manuscript, P3L9). Within this radius of 500 m around the site, there are mainly pedestrian streets parks, hills. This is the reason why the site can be considered as an urban background and not an urban traffic site.

**2.2 Measurements of aerosol light absorption and carbon monoxide**

**2.2.1 Aerosol light absorption and equivalent black carbon**

- Page 3, lines 18-26: it may be worthy to indicate more clearly this data correction procedure was applied to AE42 (and AE31) datasets only.
  - A5: Added in the revised manuscript, P3L20, L35

- Also, what could be the uncertainties related to the use of f_ values that weren't estimated for this individual instrument /site?
- For compensation parameter f values given in Drinovec et al. (2015) were used. But Drinovec et al. describe that filter loading effects change with location and time. The values of Drinovec differ a lot from Sandradewi et al (2008) (reference in manuscript but not listed in References, make sure to find the correct paper) and Zotter et al (2017). The latter paper, that is known to the authors confirms the importance of proper compensation. I would like to see a more worked out compensation correction.
  - A6: We agree with both reviewers that $f$ parameter depend on location and time and that uncertainties related to the use of $f$ values should be estimated. $f$ compensation parameter is expressed as: $f = a*(1 - w0) + 1$, with $w0$, the aerosol single scattering albedo and $a$, a constant parameter varying in the range 0.82–0.88 for the different wavelengths (950–370 nm). As a result, $f$ depends mainly on the single scattering albedo. Since no simultaneous measurements of SSA were available, we chose default values based on Drinovec et al.(2015) because they were estimated for an urban environment where SSA is expected to be lower than background or remote environments characterized by aged aerosols. The $f$-values given in Drinovec et al., (2015) correspond to SSA values of about 0.75. Later measurements during the Athens smog ACTRIS JRA1 campaign indicated that wintertime SSA value at NOA exhibits an average value of 0.8±0.05. In order to estimate uncertainties related to the chosen $f$ values, absorption coefficients calculated with $f$ values taken from Drinovec et al., (2015) were compared with those using $f$ values calculated for $w0=0.8$ (see Figure 1). Differences are found to be lower than 1% .Therefore, we estimate that on average the shadowing effect was correctly accounted for and therefore did not change the values in the manuscript. The following discussion has been added on the revised manuscript:
  - P3L28-30:*"The compensation parameter $f_\lambda$ is a parameter that mainly depends on the single scattering albedo of aerosol (SSA). Because no simultaneous scattering coefficient measurements were available, $f_\lambda$ values given in Drinovec et al.(2015) for an urban site with a single scattering albedo of about 0.75 were used for loading compensation at the urban background site.."*

[Figure]

**Figure 1 Scatter plot between absorption calculated using f_values of 1.17 (ssa=0.8) and 1.203 (Drinovec et al.2015) in**
**the shadowing effect correction algorithm.**

- Finally, what could be the impact of the PM10 cut-off, compared to the PM2.5 used at the other site?
- A7: At NOA, TSP were collected and not PM10 as indicated in the original version of the manuscript. This information has been corrected in the revised manuscript (P3L18). As the inlet includes curved tubing, a significant aerosol loss of the coarse fraction is expected. However, as indicated in several studies, BC is mainly related to fine particles
  - *P3L17-20: "AE42 was used and aerosol sampling was performed with no size-cut. As BC mainly contributes to $PM_1$ (Laborde et al., 2013; Wang et al., 2015), differences in the eBC concentrations due to the different aerodynamic diameters of sampled aerosols are expected to be negligible".*

- About the instruments At DEM (AE33) at NOA(AE42). I understand that AE33 and AE31 are compared. At page 3 line29: 'and data from AE31 aethalometer with : :: : :to AE-42, which operates continuously in parallel with the AE33 at DEM station. So what instruments are compared and where is the AE-31 located?
  - A8: Since we could not compare directly AE33 and AE42, we decided to compare AE33 with an AE31-Aethalometer (with identical measurement settings to the AE42). Both instruments were running simultaneously from 1st August to 30th September 2014 at NCSR Demokritos station. We modified the following sentence for more clarity:
  - P3L35 *"The results indicated a very good agreement between the absorption measurements (Mm$^{-1}$) from the **AE33 and AE31** instruments after compensation, with $R^2$=0.79, a small intercept of -0.15 and a slope of 0.97."*

- The R-squared of 0.79 is not very convincing for aethalometers, I would like to see the plot. The intercept (what is the unit? Inverse Mm or ng/m3?) of 0.15 is interesting.
  - A9: The plot is shown below. The unit is Mm$^{-1}$ and this information has been added in the revised manuscript. The intercept is **-0.15.** Please indicate if necessary to include this plot to the supplementary information

[Figure]

**Figure 2 Linear regression between absorption coefficients at 880 nm measured simultaneously by AE33 and AE31 and**
**corrected from loading and scattering effects using the dual Spot Technolgy and the Weingartner procedure**
**respectively, from 1st August to 30th September 2014 at NCSR Demokritos station.**

- Page 4, line 6-7: please indicate whether this value was also obtained using the 1.64 "ACTRIS correction factor" (as used by Zanatta et al., 2016)?
  - A10: As suggested by the reviewer, in the revised version of this manuscript the default C-factor for both aethalometers has been corrected with an additional correction factor. The need to use a compensation factor on top of the default value has also been confirmed by parallel measurements performed between AE31 and AE33 and a multi-angle absorption photometer (MAAP) (Model 5012, Thermo Electron Group,Waltham, MA,

USA) during 2011 at the DEM station (Diapouli et al., 2017). Absorption measurements have therefore been corrected with a factor of 1.64 as used in the ACTRIS community and proposed by the reviewer. This additional correction of our absorption measurements changes the MAC value derived from the intercomparison with EC measurements. The new corrected MAC at 880 nm value is therefore 4.6. However, correction on absorption coefficient and MAC compensate one another, and therefore final eBC values remain unchanged. The following changes have been made on the revised manuscript:

- o P3L26"*The value of 3.5 was used for $C_0$ as recommended in [...]P4L2 eBC mass concentration was derived in this study by multiplying the $b_{abs}$ coefficient at 880 nm with a constant value of mass absorption cross-section (MAC) of 4.6 $m^2$ $g^{-1}$ (determined from the comparison with simultaneous measurements at DEM of elemental carbon)*" [...] *P4L4-8: :Assuming an absorption Ångstrom exponent of 1.0, the MAC used here is 6.13 when adjusted to 637 nm. Our MAC value is at the lower limit of the values reported by Zanatta et al., (2016), for nine rural background stations across Europe (7.5-13.3 m2 g-1, calculated for 637 nm), and within the range of values reported by Hitzenberger et al., (2006) for an urban background site in Vienna (5.9 -7.5 at 637 nm).*"

- I'm not sure, if it is interesting that 'BC is historically defined from aethalometer measurements at 880nm'. The important message should be that the whole spectral de- pendence approach depends on fixation somewhere. This is done at 880nm because it is believed that at that wavelength the MAC for wood burning and fossil fuel combustion is very close. Otherwise the DEC MAC cannot be applied at NOA. The whole fractitioning is based on the wavelength dependence that is somewhere fixed (Equation 10). The reader should be convinced of the choice that is supported by literature.
  - o A11: The following sentence was added in the revised manuscript:
  - o P3L40: "*At 880 nm, no significant difference in MAC at 880 nm between eBC originating from traffic or wood-burning emission is expected  (Zotter et al., 2017)*"

- It is written '(MAC): : :. (determined from the comparison with concurrent measurements at DEM of elemental carbon)'. A bit later a reference to Diapouli et al. (2014) is included. Does this paper include the 7.5 m2 g-1? .
  - o A12: The paper of Diapouli et al. (2014) does not include absorption measurements, therefore no MAC value is presented. As indicated in the manuscript, the paper of Diapouli et al. (2014) includes an extensive description of EC/OC measurements at DEM.

- The angstrom exponent for absorption is measured why do the authors assume an exponent of 1.0 in line 6 (p4)?
  - o A13: As the reviewer mentions the absorption exponent is measured for both sites and exhibits spatial and temporal variability, with an average value superior to 1. However, this conventional value of 1 is used only at this point of the manuscript, where the scaling of the MAC calculated in this analysis based on the 880 nm wavelength absorption values, is needed to compare with the Zannata et al. 2016 paper, where MAC values from different sites were adjusted to 637 nm assuming an absorption Ångstrom exponent of 1.0.

**2.2.2 Carbon monoxide and nitrogen oxides**
- Page 4-5, BC and CO source apportionment: please discuss here possible interference from coal combustion emissions
  - o A14: Added in the manuscript, P5L18: "*It should also be noted that coal-burning organic aerosol is known to significantly absorb light at near UV wavelengths (Yang et al., 2009) and may thus interfere with $b_{abs}(\lambda_{UV})_{wb}$. Lignite coal is the single most important local energy source in Greece (Kavouridis, 2008). However, interferences from coal use are expected to be very low, as the lignite-fired power plants are located far away from Athens (>200 km distance).*"

**2.3 Source apportionment of black carbon from fossil fuel and wood burning combustion.**
- P4 line21 Reference to Sandradewi (please include the correct one in references).
  - o A15: Added

- Page 5, BC source apportionment: please justify/discuss a bit more the choice of _wb = 2 by comparison with value recently proposed by Zotter et al., 2017.
- Sandradewi discussed different Angstrom exponents depending on the chosen wavelengths. This wavelength dependence should be discussed in light of the choices given in line 32, or refer to other studies that use same wavelengths. The 470 nm channel was broken in that Sandradewi study, why does this study start at 470 nm (line 32) P5 top para. Exponents 0.9 (traffic) and 2.0 (wood) 'were used, based on the range of values.. reported'. The value of 2.0 is disqualified by Zotter et al., 2017, because is leads to differences with radiocarbon results. The exponents are crucial to the method,'based on' should be worked out.
  - o A16: In a previous study using the aethalometer model at DEM (Diapouli et al., 2017), calculations were made for different values of $a_{wb}$, in the range 1.1–3.0, by a step of 0.1. In order to identify an acceptable range of values for $a_{wb}$, the calculated babs(950)ff were correlated with NOx data, which are mainly related to fossil fuel combustion emissions. Values of $a_{wb}$ below 1.7 produced either no correlation or weak correlations and were therefore not considered acceptable values (Pearson coefficients below 0.7). On top of that, during fire events, values of angstrom exponent up to 2 have been observed at DEM (Figure 3). We expect therefore that angstrom exponent from biomass burning to be at least as high as 2. In view of these results, $a_{wb}$=2 has been selected for the study. Discussion about the choice of $a_{wb}$ has been added in the revised manuscript.
  - o P5L7-20: "*The application of the model requires the selection of suitable Ångström exponents for fossil fuel ($\alpha_{ff}$) and wood burning ($\alpha_{wb}$), since one of the greatest uncertainties of the model is associated with the a priori assumed $\alpha$ values for both types of emissions. Reported Ångström exponents range between 0.8-1.1 for pure traffic. For wood burning a wider range of values has been observed (0.9-3.5), even though αwb equal to 2.0 has long been considered a typical value for wood burning aerosol (Favez et al., 2009; Fuller et al., 2014;*

*Herich et al., 2011; Petit et al., 2014; Sciare et al., 2011). Recently, Zotter et al., (2017) recommended to use $\alpha_{ff}$=0.9 and $\alpha_{wb}$=1.68, obtained by fitting the model outputs (calculated with the absorption coefficients at 470 and 950 nm) against the fossil fraction of EC derived from 14C measurements. At DEM site, a previous study showed that values of awb below 1.7 were not appropriate for the specific site (Diapouli et al., 2017). On top of that, during fire events, Ångström exponent values up to 2 have been observed at DEM. Taking into consideration these results, absorption Ångström exponents of 0.9 and 2.0 for pure traffic ($\alpha_{ff}$) and wood burning ($\alpha_{wb}$), respectively, were used.in this study"*

     ○ *P4L36-39 "Even though different pairs of near-UV and near-IR wavelengths can be used, it is recommended to use the pair 470 nm versus 950 nm. The choice of 470 against 370 is even more critical as explained in Zotter et al (2017) since interference of VOCs or other absorbing non-BC particles can interfere with measurements with the 370 nm channel of aethalometer."*

[Figure]

**Figure 3** eBC and angstrom exponent measured at DEM station during summer forest fires.

**2.4 Source apportionment of carbon monoxide from fossil fuel and wood burning combustion**
**2.4.2 Model 2: CO BCwb BCff multiple linear regression model          6**

- P6 line 20 'the hypothesis of negligible photochemical chemistry is validated.' Where is it  validated please include reference(s). Negligible what does that mean, negligible for the scale considered in this study? Or is the assumption that BC and CO have similar lifetimes? This para needs to be worked out to convince the reader that model2 is superior to model 1.
  - ○ A17, P6L36 "The sentence was corrected to "*the hypothesis of negligible photochemical chemistry is met for the time scale considered in this study*"

**3. Results and Discussion**
**3.1 Levels and diurnal variations of black carbon and carbon monoxide**

- Page 6, line 36 - Page 7, line 7: the expected diurnal cycle of the intensity of emissions could be discussed more deeply here.
  - ○ A18: Added in the manuscript P7L14: *Based on traffic volume data (Grivas et al., 2012), a first peak in the emissions from transportation is expected around 08:00 when people commute to work, followed by a plateau from 08:00-18:00, and a secondary peak until 21:00, after when traffic is decreasing. Wood burning emissions from residential heating are expected to increase during the evening, when temperatures drop and people are back-home.*

**3.2 Source apportionment of BC and diurnal variability          7**
- Page 7 line 9 'eBC in PM2.5' but at NOA a PM10 sampling head is installed, right?
  - ○ A19: This information has been corrected on the revised manuscript.

- Page7 line 11. Apparently the 1.25 percentile of a dataset can be used for background. Really would like to read that paper. Please include Kondo et al., 2006 in the references

- o   A20: p7L32,We have added the reference to the paper of (Kondo et al., 2006), that investigated temporal variations of elemental carbon in Tokyo. The same approach was used in (Verma et al., 2010) for determining the BC background concentration  of black carbon in Guangzhou, China.

- P7L14 'relatively short lifetime of BC' please compare to P6L20
  - o   A21: We agree that there was a contradiction between statement in P7L14 and P6L20. In P6L20 we wanted to point out that both CO and BC are not chemically reactive, whereas in P7L14 we were referring to deposition losses. In order to avoid any misunderstanding we removed the sentence "*relative short lifetime*" in p7 l14.

- P7L34 how is the relative standard deviation defined in this case.
  - o   A22: It is defined as the interval of confidence in the coefficient values (slope and intercept) of the linear regression as calculated by Igor Pro software. The values are automatically calculated for each fitting.

- P7L39 '0.184' please include units if appropriate
  - o   A23: P8L19: Added

**3.3.1 Using BCwb and BCff as tracers of fossil fuel and wood burning sources        7**

- P8L4 the value 0.00137 is 0.7% of the best estimate 0.184. This is very small compared to values in Table 2. Please include discussion.
  - o   A24: 0.00137 is the standard deviation of the r'ff coefficient given from the multiple regression fitting for DEM station. This is very small compared to the range of values found in the literature (and given in Table 2) that have been calculated using different methodologies and for different type of transport fleet (size distribution and age of vehicle fleet, fuel consumption, environmental performance etc.). We assume that DEM and NOA experience similar vehicle fleet mix, and this is the reason why, the regression model was run for NOA using a fixed r'ff according to values found in DEM station (and not based on the literature, P8L22). As a result, the error estimation of r'ff was made based on the standard deviation of the model for DEM, and not on values from the literature.

- Why is 0.00137 an useable value and why is the resulting uncertainty of 25% for the emission ratio 'rather reasonable' (not scientific terminology)
  - o   A25: 0.00137 was the uncertainty calculated for DEM station. When fixing r'ff at NOA, we did a sensitivity analysis using r'ff values ranging from the lowest (0.184-0.00137) to the highest (0.184+0.00137) around the determined *r'ff*, and the resulting r'wb variated as much as 25%. We corrected the uncertainty in eq.21 and removed the statement "can be considered rather reasonable" from the revised manuscript.

- P8L13 background or intercept values are 109 and 147 how do these differences related to 'cannot differ significantly' line 1 of this page?
  - o   A26: Considering a uniform/similar vehicle fleet in terms of type of vehicle and driving patterns, the **emission ratio** between BC/CO from traffic should not differ. However, absolute concentrations of CO vary between the urban and suburban sites.

- P8L14 background concentrations of CO with a reference to Goldstein and Schade (2000), this work contains some informations on background but not on CO. Howshould the reader interpret the reference, please modify.
  - o   A27: The reference was about background estimation using regression's intercept values. In order to avoid any confusion, we removed the reference

- P8L14etc The resulting background concentration are in very good agreement with : : :1.25 percentile. Really want to learn more. For me it sounds like abracadabra.
  - o   A28: The estimation of background using the 1.25 percentile of an extended long term dataset of atmospheric concentration has been documented in previous studies with good results (Kondo et al., 2006; Verma et al., 2010). It is based on the fact that within the observed variability the minimum values of the dataset represent what we can consider background levels for this environment. It is meaningful to consider here.

- Page 8, line 25 (% COwb): please discuss these percentage regarding previous studies/results.
  A29: While there have been numerous studies in the last years investigating the contribution of different sources to black carbon surface concentrations, similar studies are very limited for carbon monoxide. Saurer et al., (2009) used the stable isotope composition of CO ($\delta^{13}C$ and $\delta^{18}O$) for the characterization of different CO sources at 3 sites in Switzerland during winter (along with other indicators for traffic and wood combustion such as $NO_x$-concentration and aerosol light absorption at different wavelengths) and estimated the wood burning contribution to night-time CO concentrations at 70%, 49% and 29% for a village site dominated by domestic heating, a site close to a motorway and a rural site respectively.  These differences reflect the spatial variability in the wood burning use within the same region depending on the type of site, as well as between countries depending on the country and the heating practices. This discussion is added in the revised manuscript (P9L13-18).

**3.3.2 Comparison the CO BCwb BCff linear model vs the CO NOx linear model        8**

- P8L41 'using a best fit line' If this is a fit how was the data selected? This was not clear from the references literature.
  - o   A30: In the reference literature there are no explanation on the methodology used to draw these two slopes. In this study, in order to o draw the minimum and maximum slopes, the 10th percentile and 90th percentile of (CO-CObgd)/NOx ratio have been calculated. To draw the minimum slope, fitting was applied for data where

CO/NOx ratio was below the calculated 10[th] percentile. To draw the maximum slope, fitting was applied to data where CO/NO$_X$ ratio was above the calculated 90[th] percentile. However, these fitted lines are just indicative of the expected range of values of CO/NOX ratios for each emission source. This information has been added in the revised manuscript (P9L29).

- P9Line4 informs us that the ratio is larger than : : :please explain
  - A31: Wood burning lines from figure 11, exhibit slopes of 20 and 25. However we do not expect to have 100% contribution of wood burning at any time of the day. We can therefore estimate that r$_{wb}$ is superior to both these values. Modifications in the manuscript: "*Nevertheless, based on "wood burning" lines from Fig. 11, and assuming that emission ratios from wood burning are similar between NOA and DEM, we estimate a r$_{wb}$ ratio for the area of Athens, larger than 25 ppbv ppbv 1.*"

- P9L9 'values found in the literature' please include references
  - A32: We removed the sentence 'values found in the literature'. The references are given in the following sentence.

- P9L10 2-3% where should I look to see the supporting material?
  - A33: An additional column in Table 4 with the COwb% resulting from the sensitivity analysis test has been added in the revised manuscript.

**4. Conclusion**
- Page 9, line 25-26: here, it sounds like wind speed is controlling the diurnal patterns. Please consider rephrasing this sentence.
  - A34: Modified in the manuscript: *"Both BC and CO displayed a clear bimodal diurnal pattern, in which morning peaks were observed due to morning inversion and rush-hour traffic, while evening peaks were attributed to combustion sources (evening traffic rush-hour, residential heating) combined with the effects of a shallow nocturnal boundary layer. Highest concentrations were observed during low wind speeds, suggesting that both combustion products were not related to regional transport but instead originated from sources within Athens."*

**Acknowledgements**
**References**

- Figure axes: please homogenize the use of "BC" / "eBC".
  - A35: Corrected

- Figure 7, right panel: legend of the y-axis seems inaccurate
  - A36: Corrected

- Table 4 Regression Slope between model 1 and model 2: what model outcomes are regressed? Are we looking at COwb/totalCO?
  - A37: Yes, this information is added in the Table

- Typos-suggestions P3 line26 'this purpose' ! for loading compensation (corrected), P4 L9 ratios were (corrected) , P4 line 25 lambda is bold (corrected) in equation P4 eq 5 lambda1 should be lambda2 in Denominator (corrected). P7L40 last ff should be sub (corrected), P8L28 diurnal variabilities : : :are (corrected),Comparison of A and B Figure 7 caption or axis titles are wrong for right bottom figure (corrected)

[revised manuscript text omitted]

---

## Editor Decision (ED1)

**Editor's technical comments on the revised version of ms. acp-2017-854 entitled "Assessment of wood burning versus fossil fuel contribution to wintertime black carbon and carbon monoxide concentrations in Athens, Greece", by A.-C. Kalogridis et al.**

François Dulac, 02 April 2018

Thank you for your revision of the manuscript. I would like you to make a few technical corrections as listed hereafter (proposed changes in your text are underlined). I will then send back your revised manuscript to the two referees.

-page 1: I had already mentioned about your initial version that the first references in your introduction are missing in your bibliography; this includes: Chaloulakou et al., 2005; Eleftheriadis et al., 1998, 2014; Kalabokas et al., 1999; Theodosi et al., 2011 (lines 30-31), Cohen et al., 2004 (line 32) and Ostro et al., 2015 (line 36).

-p.2, line 9: "observations of $NO_2$ and $SO_2$ over Athens".

-p.2, l.13: "use of wood burning observed"; "emissions from incomplete".

-p.2, l.14, p.3, l.29: insert a space between "al." and the year within parentheses.

-p.2, l.15: "increase in".

-p.2, l.18: I do not understand the use of "Respectively, " here: remove it?

-p.2, l.14; p.3, l.27; p.4, lines 6 and 7; p.5, l.11; p.9, l.14: remove the comma before the year within parentheses.

-p.2, l.21: you should cite Diapouli et al., 2017a before 2017b; interchange references and chance citations accordingly.

-p.2, l.27: "between 10 and 2000".

-p.3, l.18: the reference Wang et al. (2015) is missing in your bibliography.

-p.3, l.26: insert a space before "The".

-p.3 l.30: I feel necessary here to add information about the sensitivity test on the value of $f$ that you mention in your reply to referees.

-p.4, lines 1-3: you should shift the significance of "MAC" at its first occurrence rather than at the second one.

-p.4, l.3: a space is missing before the opening parenthesis.

-p.4, l.18: remove the comma after "1-min".

-p.4, l.20, and p.9, l.18: insert a comma before " respectively".

-p.4, l.21: remove the dot at the end of the section title.

-p.4, l.28: insert " ($\lambda$)" after "wavelength".

-p.4, l.29: move "$b_{abs}$": "of light by aerosols ($b_{abs}$) is".

-p.4, l.39: "the 370-nm aethalometer channel".

-p.5, l.13: use the exponent style in "$^{14}$C".

-p.5, lines 29-30: 3 new references are missing in the bibliography.

-p.6, l.17: move the comma before rather than after "$NO_x$".

-p.6, l.26: "are the relevant".

-p.6, l.29: "applied to".

-p.7, l.17: "back home" without an hyphen.

-p.7, l.29: "and an increase".

-p.8, l.1: "suppression" does not seem appropriate; I suggest "the decrease in".

-p.8, l.2: insert ":" after "scales".

-p.9, l.19: I suggest "depending on the regional heating practices"..

-p.9, l.30: "the lowest".

-p.9, l.38: remove the article after "varying".

-p.10, lines 2 and 3: "by a factor of".

-p.10, l.6: "that the $CO$-$NO_x$ linear model".

-p.10, l.22: "The highest".

-p.10, Acknowledgements: "loan of an aethalometer. We also acknowledge".

-Fig. 3: remove the letters added in parentheses after city labels in the plot (or use them to list the references in the legend); insert a space after "Barcelona".

-Fig. 4: add "(left)", "(right)", "(blue)", and "(red)" after "BC", "CO", "NOA", and "DEM"; specify what vertical bars do represent in this plot.

-Fig. 10: explain in the legend what are the box limits, vertical and horizontal bars, and coloured dots.

-Fig. 11: clarify in the figure legend what are the 3 plotted lines.

-I suggest that Fig.2 in your reply is added as supplementary information.

---

## Author Response (AR2)

**As requested by the editor, we made all the technical corrections requested. All relevant changes made in the manuscript are highlighted in yellow in this version of the manuscript.**

[revised manuscript text omitted]

---

## Editor Decision (ED2)

**Editor's comments on the June 2018 revised version of ms. acp-2017-854 entitled "Assessment of wood burning versus fossil fuel contribution to wintertime black carbon and carbon monoxide concentrations in Athens, Greece", by A.-C. Kalogridis et al.**

François Dulac, 15 June 2018

Thank you for your further revision of the manuscript. I am pleased to accept it for publication pending a few technical corrections as listed hereafter:

-Section 2.2.1: please revise the notation of parameters related to equation 3; in the equation itself, the parameter $a$ should read $\alpha$; I recommend using "$f_\lambda$" and "$SSA_\lambda$" in the equation and in the text before the equation or after (replace the "$i$" by "$\lambda$" or by respective $\lambda$ values); the variable SSA should also be in italic style.

-Section 2.3 and 2.4: equations 11 to 16 are presently erroneously numbered from 3 to 8; please also check possible references to those equation numbers in your text.

-References: add a final dot in the ref. of Diapouli et al., 2017b.

-Table 1: "NO$_x$ (DEM)*" and "NO$_x$ (NOA)" (move the star sign and use the subscript style for x.

-Legend of figure 2: please specify "at NOA (in blue) and DEM (red)".

-Legend of figure 10: please specify "of CO$_{ff}$ (left) and CO$_{wb}$ (right) at DEM (top) and NOA (bottom) stations".

---

## Author Response (AR3)

**acp-2017-854    Submitted on 11 Sep 2017**
**Assessment of wood burning versus fossil fuel contribution to wintertime Black Carbon and Carbon monoxide concentrations in Athens, Greece**

**01/06/2018**

**Author's response**

The authors acknowledge both referees for critically reading the manuscript and for their contribution in improving and clarifying this study. This document includes a point-by-point response to the reviews, followed by a marked-up manuscript version

**Referee 1**

1) **Rephrase P.1 lines 18-20 and P.2, lines 38-40, so that readers doesn't feel like authors are actually using the CO/NOx ratio approach to estimate contributions.**

   a. P.1 lines 18-20 We replaced *independent evaluation* by *comparative analysis* so that it is not implied that a higher credit is given to the CO/NOx model

   b. P.2, lines 38-40 "*A new approach based on the relations between CO, $BC_{wb}$ and $BC_{ff}$ was used for the source apportionment of CO, and compared with a method based on the $CO/NO_x$ emission ratios.*"

**2) Make use of the comparison between AE31 vs. AE33 measurements at 450nm, 880nm, and 950nm to further validate the choice of the fλ values. This may be illustrated in the supplementary material (which currently only includes results obtained at 880nm, and doesn't seem to be adequately referred to in the main text).**

- We have removed the comparison of the AE31 and AE33 as suggested by the referee 2. However, we have added an additional validation of the compensation parameter to validate the choice of the fλ values The following has been added in the revised manuscript: "*The performance of the loading compensation algorithm was examined using the approach described in Drinovec et al., (2017), where the dependence of the absorption Ångström exponent α to the attenuation of light is analyzed. The slope that derives from the regression between the Ångström exponent α (calculated from α=ln(babs470 nm/b880nm)/ln(880/470)) and the attenuation has been calculated. The value of the slope is equal to -0.0055 before compensation and close to 0 (-0.0005) after compensation of the data, thus indicating that the shadowing effect was correctly accounted for.*"

**Referee 2: General**

**The Manuscript attributes the concentration-contribution of fossil fuel and wood burning to air pollution in Athens and with that handles an important health related subject that needs attention. For the source apportionment of black carbon, the well-known technique based on wavelength dependence of the aerosol light absorption coefficient is used after application of necessary compensations. The paper consciously refers to recent work. The aethalometer-model work adds a new estimate of woodburning contribution to BC. The CO source apportioning (model 2) is a new application and it is interesting to see how this will be followed up. Model 2 combined with the aethalometer model could be used as an independent estimate of bottom up emission estimates.**

- **An important aspect in the manuscript is the conversion from deduced absorption coefficients to mass concentrations by using the MAC at 880 nm. In this paper a constant MAC (page 4line7) of 4.6 m2 g-1 is used, in the paper by Diapouli et al (2017b), from the same group using the same site and instruments, the manuscript under review is referenced with a MAC of 7.5 m2 g-1; "Kalogridis et al. calculated a MAC of 7.5 m2 g-1 for the DEM station......". Using independent EC measurements, Diapouli found a MAC of 4.1, they however used for the compensation a C value of 3 instead of 3.5 (p3l26).-☐ When calculating: 4.1 x3.5/3=4.8 m2 g-1 a value very close to 4.6 m2 g-1 is obtained. I understand the history of the different MAC values references in Diapouli and in this manuscript, but it should be clear to other reader too.**

  - It is appreciated that the new metrics and data procedure in the manuscript is acknowledged by the referee. The new MAC value (4.6) presented in this paper (following revisions of the first version) is now much closer to the value found in the study of Diapouli et al. (2017). As pointed out by the referee, the reference to a MAC value of 7.5 is outdated, since it was cited prior to the revisions made in the present paper. In order to not confuse the reader and to be consistent in the MAC values presented for DEM station, the MAC value of 7.5 will be changed to 4.6 in the paper of Diapouli et al. (2017), with an introduction of a corrigendum as soon as the present paper is accepted and published in ACP (in order to include the correct doi in the reference)..

- W.r.t. Model 2. For the source identification of CO the emission ratio of fossil fuel is determined by regression for DEM. For NOA this value is adopted. The 0.184 ppbv ng$^{-1}$ m$^3$, is a value I cannot criticize since little reference to other sources is given (Table 2 includes some values). The error 0.00137 ppbv ng-1 m3 seems to be very small **and I assume that it stands model for the precision and not for the accuracy, the spread or variability is certainly much larger. When applying this error in a sensitivity calculation (page 9 top) I believed not much is to be learned. The exercise however lead to a uncertainty estimate of 25% for the woodburning ratio, that seems to be a very large value as compared to 0.00137 out of 0.184.**

  - We do agree with the above assessment of the reviewer. It is now made clear in the manuscript that the error range selected for the sensitivity analysis was chosen equal to one standard deviation of the regression analysis, and thus stands for the model precision: P8L14:" *the determined regression coefficients ($r'_{ff}$ and $r'_{wb}$ for DEM) were found with a precision (standard error of regression) below 2 %*" & P8L23 "*equal to one standard deviation in the regression analysis according to equation 20*". We do think that this results suggests that the coefficients calculated here for Athens in the case of $r_{wb}$ are prone to high uncertainty (25%) but this is not unreasonable considering the variability of the wood burning sources. Finally, we provide to the reader the full information from this exercise.

  **Moreover, I think that other quickly available ratio estimates are available. From reported emissions from traffic and household woodburning, I findthe that approximately EC/CO woodburning is double EC/OC traffic (see table herebelow). This is somewhat different from the values given in Equation 19-21. Compare e.g. (my calculation) 9.5/4.3 with (manuscript)8.8/5.4. Also the values discussed in Verma (table 2) hind to slightly bigger ratio of ratios.**

| EC/CO (2014-2015) (kg/kg) | |
|---|---|
| **all traffic (mix)** | **0.0043** **0.0040** |
| **woodburning** | **0.0095** **0.0093** |

**Table: bottom up emission ratio estimates (kg/kg), please notice that the units are different from the units in the manuscript): EC (mass C in PM2.5)/CO (mass) for 'woodburning' (households only) and traffic (all road traffic! including highways, trucks)**

  - We don't have a knowledge of the source of this data and we cannot comment the range of values further.

Specific

- **P3 line25 , 'loading effect, i.e. the fact that the attenuation increases as light absorbing particles accumulate in the filter..'. loading effect is the loss of linear relationship between the attenuation and surface loading. Or give a similar correct explanation.**
  - Modified in the revised manuscript P3L25: *"the linearity loss in the relationship between the transmission of light through the sample-laden filter and the amount of the light-absorbing sample on the filter"*

- **P3line 28, 'f values come from Drinovec', please repeat the values in this paper.**
  - Values have been added in the revised manuscript

- **The values are valid for SSA=0.75, how do you know? Drinovec paper does not include information on SSA.**
  - The SSA has been estimated using the following formula: $f = a \cdot (1 - SSA) + 1$, and assuming an $\alpha$ value of 0.83. The latter formula has been added in the revised manuscript, in order that it is clearer to the reader.

- **P3 line32, 'calculated for a SSA value of 0.8', explain how:**
  - Added in the revised manuscript P3L28-31: *"The compensation parameter $f_\lambda$ is a parameter that mainly depends on the single scattering albedo of aerosol and is expressed as: $f = a \cdot (1 - SSA) + 1$ (3), where SSA is the aerosol single scattering albedo and $\alpha$ a constant parameter, varying in the range 0.82–0.88 for the different wavelengths (950–370 nm)*
  - *"*

- **P3line32 'Differences were found to....' Differences in what?**
  - Added in the revised manuscript P3L36: *"Differences in the absorption coefficients calculated using f_values calculated for an SSA of 0.8 (e.g $f_6$=1.166) and those found in Drinovec et al., (2015) "*

- **P3 Line32 'thus it is estimated that' How is this estimated? In Drinovec it is discussed how this can be checked. E.g. plot the Angstrom exponent as a function of ATN.**
  - As suggested by the reviewer we did this extra-check in order to validate the compensation. We found after compensation a slope close to 0. The following has been added in the revised manuscript P3L33: *"The performance of the loading compensation algorithm was examined using the approach described in Drinovec et al., (2017), where the dependence of the absorption Ångström exponent $\alpha$ to the attenuation of light is analyzed. The slope that derives from the regression between the Ångström exponent $\alpha$ (calculated from $\alpha=ln(babs470\ nm/b880nm)/ln(880/470))$ and the attenuation has been calculated. The value of the slope is equal to -0.0055 before compensation and close to 0 (-0.0005) after compensation of the data, thus indicating that the shadowing effect was correctly accounted for."*

- **P3line37 'reasonable' what is reasonable? To my opinion this whole section is not necessary. I suggest to leave out L34 to P4line2**
  - As suggested by the referee the section has been removed.

- **P4 MAC see general** (see comment above)
- **P6line8 add "source" in front of apportioning** (added)
- **P6 line 37 CO and BC are exclusively** (corrected)
- **P7L25/26 of the same order .... as (instead of than)** (corrected)

- **P8L31 'a relative standard deviation below 2%' what does this imply w.r.t uncertainty, precision, and accuracy?See also my general remarks.**
  - The standar error of the regression assess the precision of the model. This has been included in the text P8L14: *"the determined regression coefficients ($r'_{ff}$ and $r'_{wb}$ for DEM) were found with a precision (standard error of regression) below 2 %"*

- **P8L38 I'm not familiar with the parameter size distribution when discussing a fleet mix but it comes to me as awkward.**
  - The "size distribution' was here refereeing to the dimension of the car. This term has been replaced by *'Vehicle size class'* in the revised manuscript.

- **P8L39 'We acknowledge.....uncertainty. The latter is difficult to be estimated with accuracy' Please rephrase."**
  Modified: *"It should be noted here that this assumption might introduce some uncertainty to the results"*

- **Figures 7 caption states: '(right) BCwb and BCff.' But axis labels have BCwb and COmeasured. I think label y axis is wrong scale is more BC-like. Please make all ranges x and y axis same. Thus e.g. BC 0-20 ng m-3 and CO 0-6000 ppb.**(Corrected)

**Figures 8 same ranges as Figure 7** (Corrected)

[revised manuscript text omitted]

$$=108.53 + 0.18414 \times BC_{ff}\,(ng.m^{-3}) + 0.1142 \times BC_{wb}(ng.m^{-3})$$

$$=146.81 + 0.18414 \times BC_{ff}\,(ng.m^{-3}) + 0.13089 \times BC_{wb}\,(ng.m^{-3})$$

**Figure 8: Best-fit linear correlations between CO and a combination of $BC_{ff}$ and $BC_{wb}$ for DEM (a) and NOA station (b).**

[Figure]

5  **Figure 9: Time series of the calculated $CO_{ff}$, $CO_{wb}$ and $CO_{bgd}$ concentration at (a) DEM and (b) NOA stations.**

[Figure]

**Figure 10: Diurnal variations of CO$_{ff}$ and CO$_{wb}$ at DEM and NOA stations.** Data are presented as box and whisker plots, where boxes encompass values between the 25th and 75th percentiles, horizontal lines represent median values, and 'whiskers' give the 80% range of the values, whereas colored markers represent the mean values.

[Figure]

**Figure 11: Scatter plot for CO and NO$_x$ data from NOA (blue) and DEM (red) along with best fit lines, aligned in the lower edge of CO versus NO$_x$ scatter charts (black), and in the upper edge (blue and red for NOA and DEM, respectively).**

---

## Author Response (AR4)

**Author's Response to Editor's comments on the revised version of ms. acp-2017-854 entitled "Assessment of wood burning versus fossil fuel contribution to wintertime black carbon and carbon monoxide concentrations in Athens, Greece", by A.-C. Kalogridis et al. 28 June 2018**

5    Dear Editor, Thank you for your further revision and help in improving the manuscript. We have corrected all the technical issues as requested. All corrections are marked-up in yellow in the following version of the manuscript.

[revised manuscript text omitted]

$= 108.53 + 0.18414 \times BC_{ff}(ng.m^{-3}) + 0.1142 \times BC_{wb}(ng.m^{-3})$

$= 146.81 + 0.18414 \times BC_{ff}(ng.m^{-3}) + 0.13089 \times BC_{wb}(ng.m^{-3})$

**Figure 8: Best-fit linear correlations between CO and a combination of BC$_{ff}$ and BC$_{wb}$ for DEM (a) and NOA station (b).**

[Figure]

5    **Figure 9: Time series of the calculated CO$_{ff}$, CO$_{wb}$ and CO$_{bgd}$ concentration at (a) DEM and (b) NOA stations.**

[Figure]

**Figure 10: Diurnal variations of CO$_{ff}$ (left) and CO$_{wb}$ (right) at DEM (top) and NOA (bottom) stations.** Data are presented as box and whisker plots, where boxes encompass values between the 25th and 75th percentiles, horizontal lines represent median values, and 'whiskers' give the 80% range of the values, whereas colored markers represent the mean values.

[Figure]

**Figure 11: Scatter plot for CO and NO$_x$ data from NOA (blue) and DEM (red) along with best fit lines, aligned in the lower edge of CO versus NO$_x$ scatter charts (black), and in the upper edge (blue and red for NOA and DEM, respectively).**